# Role of ammonia in European air quality with changing land and ship emissions between 1990 and 2030

Sebnem Aksoyoglu[1], Jianhui Jiang[1], Giancarlo Ciarelli[2], Urs Baltensperger[1], André S. H. Prévôt[1]

5  [1]Laboratory of Atmospheric Chemistry, Paul Scherrer Institute, 5232 Villigen PSI, Switzerland
[2]Institute for Atmospheric and Earth System Research/Physics, Faculty of Science, University of Helsinki, Finland

*Correspondence to*: Sebnem Aksoyoglu (sebnem.aksoyoglu@psi.ch) and Jianhui Jiang (jianhui.jiang@psi.ch)

**Abstract.** The focus of this modeling study is on the role of ammonia in European air quality in the past as well as in the future. Ammonia emissions have not decreased as much as the other secondary inorganic aerosol (SIA) precursors nitrogen oxides ($NO_x$) and sulfur dioxide ($SO_2$) since 1990s, and are still posing problems for air quality and the environment. In this study, air quality simulations were performed with a regional chemical transport model at decadal intervals between 1990 and 2030 to understand the changes in the chemical species associated with SIA under varying land and ship emissions. We analyzed the changes in air concentrations of ammonia, nitric acid, ammonium, particulate nitrate and sulfate as well as changes in the dry and wet deposition of ammonia and ammonium. The results show that the approximately 40% decrease in SIA concentrations between 1990 and 2010 was mainly due to reductions of $NO_x$ and $SO_2$ emissions. The ammonia concentrations on the other hand decreased only near the high emission areas such as the Netherlands and northern Italy by about 30% while there was a slight increase in other parts of Europe. Larger changes in concentrations occurred mostly during the first period (1990-2000). The model results indicate a transition period after 2000 for the composition of secondary inorganic aerosols due to a larger decrease in sulfate concentrations than nitrate. Changes between 2010 and 2030 - assuming the current legislation (CLE) scenario - are predicted to be smaller than those achieved earlier for all species analyzed in this study. The scenario simulations suggest that if ship emissions will be regulated more strictly in the future, SIA formation will decrease especially around the Benelux area, North Sea, Baltic Sea, English Channel and the Mediterranean region, leaving more ammonia in the gas phase which would lead to an increase in dry deposition. In the North, the decrease in SIA would be mainly due to reduced formation of particulate nitrate while the change around the Mediterranean would be caused mainly by decreased sulfate aerosol concentrations. One should also keep in mind that potentially higher temperatures in the future might increase the evaporation of ammonium nitrate to form its gaseous components $NH_3$ and $HNO_3$. Sensitivity tests with reduced $NO_x$ and $NH_3$ emissions indicate a shift in the sensitivity of aerosol formation from $NH_3$ towards $NO_x$ emissions between 1990 and 2030 in most of Europe except the eastern part of the model domain.

## 1 Introduction

Ammonia ($NH_3$) plays an important role in atmospheric chemistry. As an alkaline gas, it affects the acidity of clouds and precipitation and it is one of the main sources of reactive nitrogen (Simpson et al., 2011; Fowler et al., 2015). Studies show that ammonia emissions not only are toxic for plants and lead to a loss of biodiversity (Jones et al., 2014; Roth et al., 2015), but they also contribute significantly to the formation of particulate matter (Fowler et al., 2009; 2015). Ammonia reacts very rapidly with sulfuric acid ($H_2SO_4$), which is formed from the oxidation of $SO_2$ by OH in the gas phase and by $O_3$, hydrogen peroxide ($H_2O_2$) and other oxidants in the aqueous phase, to form ammonium sulfate (($NH_4$)$_2SO_4$) or ammonium bisulfate ($NH_4HSO_4$) (Seinfeld and Pandis, 2012). If there is enough ammonia available after the neutralization of $H_2SO_4$, it reacts with nitric acid ($HNO_3$) to produce ammonium nitrate. These secondary inorganic aerosols (SIA) contribute most to the fine particulate matter ($PM_{2.5}$) in Europe (Ciarelli et al., 2016; 2019; Aksoyoglu et al, 2017). Although ammonia and ammonium are nutrients for plants and are used as fertilizers, they are the largest contributors to nitrogen pollution of ecosystems through eutrophication and acidification (Jones et al., 2014). The main sources of ammonia emissions are agricultural,

including volatilization of animal waste and synthetic fertilizers but a small fraction ($< 10\%$) comes also from other sources such as industry, household and traffic (UNECE, 2019).

European anthropogenic emissions have decreased substantially since the 1990s as a result of large emission reductions following the Gothenburg Protocol (GP) (UNECE, 1999), revised Gothenburg Protocol (revised on 4 May 2012, https://www.unece.org/env/lrtap/multi_h1.html) and EU Directives (https://www.eea.europa.eu/data-and-maps/indicators/main-anthropogenic-air-pollutant-emissions/assessment-6). Several studies investigated the effects of reduced land emissions on the air quality in various parts of Europe (Guerreiro et al., 2014, Aksoyoglu et al., 2014; Wichink Kruit et al., 2017; van Zanten et al., 2017; Theobald et al., 2019; Ciarelli et al., 2019). The largest decrease was in $SO_2$ emissions (by more than 90% in 2017 compared to 1990), followed by $NO_x$ and NMVOC (non-methane volatile organic compounds) emission reductions (more than 50%), while ammonia emissions decreased less – approximately 23% on average in the EU-28 countries. Ammonia emissions have been increasing again since 2014, however, posing problems for Europe (NEC, 2019). This is mainly due to the difficulty in implementing additional emission reductions in the agriculture sector, especially in the housing of animals and the storage and application of animal manures. The large decrease in sulfur emissions over the last few decades has changed the aerosol composition: particulate nitrogen was dominated by sulfates in the 1990s while today nitrate predominates (Colette et al., 2016).

Recent studies showed that the decline in nitrogen deposition in the past was mainly due to the decreased deposition of oxidized nitrogen components as a consequence of large emission reductions in Europe. Deposition of reduced nitrogen (ammonia $NH_3$, particulate ammonium $PNH_4$) was predicted to increase further in the future (Aksoyoglu et al., 2014; Simpson et al., 2014; Colette et al., 2016).

While land emissions have been significantly reduced over the last few decades, emissions from the least regulated sector, maritime transport, have been increasing (Jonson et al., 2015). The International Maritime Organization (IMO) controls ship emissions globally through the Marine Pollution Convention (MARPOL) Annex VI, which limits the main air pollutants contained in ship exhaust gas and prohibits deliberate emissions of ozone depleting substances (ODS) (http://www.imo.org/OurWork/Environment/PollutionPrevention/Pages/Default.aspx). The revised MARPOL Annex VI with the aim of significantly strengthening the emission limits entered into force on 1 July 2010. In addition, emission control areas (ECAs) were introduced to reduce emissions further in designated sea areas. For example, in Europe, the North Sea and Baltic Sea areas were defined as SECAs (sulfur emission control areas), where the limits were restricted to 1.0% in July 2010 and further reduced to 0.1% in 1 January 2015. New global sulfur emission regulations, which reduce limits from 3.5% to 0.5% came into force on 1 January 2020 (https://www.imo.org/en/MediaCentre/HotTopics/Pages/Sulphur-2020.aspx, last access on 23.10.2020). On the other hand, there has been an increase in the emissions of other species, especially $NO_x$, in all European sea areas (Colette et al., 2016). The nitrogen emission control area (NECA) around the North Sea, Baltic Sea and the English Channel will enter into force in 2021 but only for newly-built ships (EEA, 2013). $NO_x$ emissions from all existing and new ships outside the NECA areas will continue to be under-regulated.

Based on recent studies, ship emissions are considered to be a major source of air pollution especially around the coastal areas of Europe (Jonson et al., 2019; Pay et al., 2019; Toscana and Murena, 2019). According to the European Environment Agency, emissions of nitrogen oxides from international maritime transport in European waters are projected to increase and could be equal to land-based sources by 2020 (EEA, 2013). Viana et al. (2014) reviewed a series of studies performed before 2012 dealing with the impact of shipping emissions on air quality

in the European coastal areas and reported that contribution of ship emissions to $PM_{2.5}$ and to $NO_2$ vary between 1-14% and 7-24%, respectively, depending on location and time. In a recent model-intercomparison study, Karl et al. (2019) evaluated the contribution of ship emissions to air quality in the Baltic Sea region in 2012 to investigate the differences among model predictions and showed that variations in ship-related $PM_{2.5}$ were mainly due to differences in the models' schemes for inorganic aerosol formation. Another study reported a contribution of 45% to $PM_{2.5}$ concentrations by ship emissions in the Mediterranean and 10-15% around the Baltic Sea and concluded that the evolution of $NO_x$ emissions from ships and land-based $NH_3$ emissions would play a significant role in future European air quality (Aksoyoglu et al., 2016).

In a previous study, we investigated the changes in ozone and $PM_{2.5}$ during the period of 1990-2030 (Jiang et al., 2020). In the present paper, we focus on ammonia as an important precursor of secondary inorganic aerosols to investigate 1) how it affected the air quality in Europe between 1990 and 2010 when land emissions were reduced significantly; 2) how it will affect the air quality between 2010 and 2030 when there will also be more strict reductions in ship emissions in addition to further reductions in land emissions and 3) how the sensitivity of aerosol formation to $NO_x$ and $NH_3$ emissions would vary between 1990 and 2030.

## 2 Method

### 2.1 Air Quality Model

We performed simulations over the European domain using the regional air quality model CAMx (Comprehensive Air quality Model with eXtensions) version 6.50 (Ramboll, 2018) for 1990, 2000, 2010, 2020 and 2030. The model domain covers an area from 17°W to 39.8°E and from 32°N to 70°N with a horizontal resolution of 0.25° and 0.4°. We selected from the meteorological layers 14 terrain-following vertical layers ranging from 50 to 8000 m asl to be used in CAMx. The gas-phase chemical mechanism was Carbon Bond 6 Revision 2 (CB6r2) (Hildebrandt Ruiz and Yarwood, 2013). The fine/coarse option for the particle size was selected to calculate the aerosol concentrations in the $PM_{2.5}$ fraction. Organic aerosols were calculated using the secondary organic aerosol chemistry/partitioning (SOAP2.1) module (Ramboll, 2018) and the ISORROPIA thermodynamic model was used for the partitioning of inorganic aerosol components (Nenes et al., 1998).

### 2.2 Deposition Scheme

Dry and wet deposition of species were calculated using the Zhang scheme (Zhang et al., 2003; Ramboll, 2018). Although bi-directional air-surface exchange of $NH_3$ has been observed over a variety of land surfaces, most of the chemical transport models (CTMs) treat this exchange only as dry deposition that might lead to an underestimation of daytime $NH_3$ concentration because of overestimated dry deposition (Zhang et al., 2010). Wichink Kruit et al. (2012) reported that the inclusion of a stomatal compensation point led to increased modeled ammonia concentrations in agricultural areas in the Netherlands. Since stomatal compensation points are affected by the canopy type, temperature, growth stage, meteorological conditions, nitrogen status and cutting practices, it is very difficult to implement it in CTMs due to imprecise knowledge about the sub-grid variations in concentration, vegetation type and fertilizer applications (Huijsmans et al., 2018; Skjoth et al., 2011). Although the introduction of such a compensation point improves the model performance, the modeling of ammonia remains challenging due to temporal and spatial variations of emissions and grid resolution (Sutton et al., 2013). The bi-

directional ammonia algorithm of Zhang et al. (2010) has been added recently as an option to the original Zhang deposition algorithm in the latest version of CAMx (v7.00). Default landuse-dependent emission potentials control ammonia compensation points along the surface-air transport circuit. When the atmospheric ammonia concentration exceeds the compensation point, the net flux is from air to surface; in the opposite case, the net flux is from surface to air. Although the Zhang dry deposition algorithm in the previous version of the CAMx (v6.50) model used in this study did not include compensation points, it did treat bi-directionality indirectly by using a deposition parameter that strongly influenced ammonia deposition via surface resistance. The surface resistance is an area of great uncertainty in deposition models. Surface wetness plays an important role for both cuticular and ground resistance. This effect is included in some deposition velocity algorithms. The parameterizations for wet cuticles and ground are quite variable between models. Some models such as CMAQ use the Henry's Law constant to account for the solubility of chemical species, the EMEP model (Simpson et al., 2012) considers the chemical content of dew by treating co-deposition of species such as $SO_2$ and $NH_3$ while Zhang et al. (2003) includes the consideration of friction velocity. Wichink Kruit et al. (2017) showed the effects of co-deposition, chemistry and meteorology during 1993-2014 in the Netherlands. Relatively wet conditions led to lower ammonia concentrations, while warm and dry conditions led to higher levels.

Wet deposition is the predominant removal process for fine particles. Particles act as cloud condensation nuclei and the resulting cloud droplets grow into precipitation. The CAMx wet deposition model employs a scavenging approach using the 3-dimensional cloud/rain input from the meteorological model. Banzhaf et al. (2012) reported that droplet pH variation within atmospheric ranges affects modeled air concentrations and wet deposition fluxes significantly. The pH-dependent parameterizations are incorporated and cloud water pH is calculated by the aqueous-phase chemistry algorithms in the CAMx model.

**2.3 Input Data**

Some of the input data used in this study were obtained from the EURODELTA-Trends (EDT) project (Colette et al., 2017). The meteorological data in the EDT project was produced by the Weather Research and Forecast Model (WRF version 3.3.1) in the EuroCordex domain with a resolution of 0.44°. We re-gridded the data to our model domain and prepared the meteorological input parameters for the CAMx model by means of the preprocessor WRFCAMx version 4.4 (http://www.camx.com/download/support-software.aspx). Another input dataset provided by the EDT project was the initial and boundary conditions which were based on monthly climatological data (Colette et al., 2017). The ozone column densities were prepared using the Total Ozone Mapping Spectrometer (TOMS) data from NASA, and photolysis rates were calculated using the Tropospheric Ultraviolet and Visible (TUV) Radiation Model version 4.8.

Anthropogenic emissions for the three base cases (1990, 2000 and 2010) were obtained from the EDT database and adjusted to the CB6r2 chemical mechanism in CAMx as described in Jiang et al. (2020). The biogenic emissions (isoprene, monoterpenes, sesquiterpenes and soil-NO) were generated using the Model of Emissions of Gases and Aerosol from Nature (MEGAN) version 2.1 (Guenther et al., 2012). The anthropogenic emissions were distributed to various vertical layers depending on their sources using the vertical profile given by Bieser et al. (2011). The ship emissions over the sea were injected into the second model layer. All the biogenic emissions were released into the surface layer.

The anthropogenic emissions for scenarios in 2020 and 2030 were prepared using the current legislations in Europe according to the revised Gothenburg Protocol (revised on 4 May 2012, https://www.unece.org /env/lrtap/multi_h1.html) and the National Emissions Ceilings (NEC) Directive (2016/2284/EU), respectively. The ship emissions in 2020 and 2030 are projected based on current legislation (CLE) of the International Maritime Organization (IMO) and EU (2020_CLE, 2030_CLE). We also carried out two more scenarios for 2020 and 2030 in which ship emissions were kept the same as in 2010 to estimate the expected changes in their contributions to air pollution. Another scenario, 2030_MTFR (maximum technically feasible emission reductions) was used to investigate the effect of maximum reductions of ship emissions (Jiang et al., 2020). In all scenarios for 2020 and 2030, meteorology and boundary conditions were kept the same as in 2010 and only anthropogenic emissions were changed.

Some additional tests were also performed to determine whether there has been any change in the sensitivity of aerosol formation to $NO_x$ and $NH_3$ emissions due to changes in the European emissions over the four decades. The simulations for 1990 and 2030_CLE were repeated with 30% reductions in $NO_x$ and $NH_3$ emissions: (1) 1990 with 70% $NO_x$, (2) 1990 with 70% $NH_3$, (3) 2030_CLE with 70% $NO_x$, (4) 2030_CLE 70% $NH_3$. The modeled SIA concentrations in each case were compared with those in the corresponding base cases, *i.e.* 1990 and 2030_CLE, respectively. A description of all simulations performed in this study is shown in Table 1.

## 3 Results and Discussion

### 3.1 Model evaluation

The model results for 1990, 2000 and 2010 were compared with the measurements available at the EDT project database which is based on EMEP datasets (https://wiki.met.no/emep/emep-experts/tfmmtrendstations). The number of available measurement stations varies between 15 and 64 depending on the year and species. For ozone, only measurements at the background-rural stations were used to reduce uncertainties due to the model resolution. Model performance for $SO_2$, $NO_2$, $PM_{10}$, $PM_{2.5}$ and hourly $O_3$ was discussed in detail in Jiang et al. (2020). In the present study, we performed additional evaluations for ammonia, total ammonium, total nitrate and secondary inorganic aerosol components. Since measurements with large spatial and temporal coverage for such species are scarce, only the 2010 measurements from the EBAS database (http://ebas.nilu.no, last access: 10 July 2020) were compared with the model results.

Atmospheric concentrations of ammonia are not well characterized due to relatively small number of monitoring sites, the short lifetime of $NH_3$ in the air and the difficulty of measuring non-point source emissions such as agricultural fields. Most of the measurement sites used in this study are located in the north; only very few stations are in the other parts of Europe (Fig. 1). The detailed information about the measurements (location, methods, temporal resolution) at each site is given in Table S1. Most of the measurements are daily concentrations, except for some sites in the Netherlands (hourly), Spain and Italy (weekly), Switzerland (bi-weekly) and UK (monthly). Measurement methods also differ; most of the stations use filter-pack sampling, while the passive samplers were used at 2 sites in Spain and the denuder systems were adopted at sites in the Netherlands, Great Britain and Switzerland. One should keep in mind that sampling artefacts due to the volatile nature of ammonium nitrate and the possible interaction with strong acids make separation of gases and particles by simple aerosol filters less reliable as indicated by EMEP (Co-operative Programme for Monitoring and Evaluation of the Long-Range

Transmission of the Air Pollutants in Europe), (https://projects.nilu.no/ccc/reports/cccr1-2019_Data_Report_2017.pdf). Modeled ammonia concentrations are similar to the measured ones at the few sites in the south while one site in eastern Europe shows an underestimation (Fig. 1). On the other hand, ammonia is overestimated at several sites such as in the UK, and in high emission areas around the Netherlands and Denmark. The mean fractional bias at all sites is 37.9% (Table S2). Overestimation might originate from either overestimated emissions or underestimated removal (deposition, particle formation). There are still large uncertainties about ammonia emissions. Recent studies show that better agreement between models and measurements can be achieved when ammonia emissions are modulated with local meteorological conditions (Backes et al., 2016; Hendriks et al., 2016). Most models, however rely on the static ammonia emission profiles provided in the emission inventories (Ciarelli et al., 2019).

Comparison of measured and modeled total ammonium (sum of gaseous ammonia and particulate ammonium) is an additional test for model evaluation. There are more stations with measurements of total ammonium and total nitrate (sum of nitric acid and particulate nitrate) than for ammonia (Table S2). Statistical parameters indicate an overestimation for total ammonium and nitrate, the variation of the mean bias among the measurement sites, however, suggests that overestimation mostly occurs around high emission areas in central Europe, while modeled and measured concentrations are similar at most of the sites, especially in the Iberian Peninsula and in Scandinavia (Fig. 1).

These results suggest that ammonia emissions in the emission inventory might be too high around the main emission sources in central Europe and/or deposition is underestimated by the model for which the resolution might also be a critical factor. The model estimate for a grid cell might not be representative of the specific location of the measurement site. Especially in mountainous areas with very spatially variable precipitation patterns, spatial representativeness of the measurement sites is not possible. Furthermore, measurement sites close to farming areas may overestimate deposition of reduced nitrogen with respect to the modeled average deposition within the grid cell. In addition, several studies show that the dry deposition velocity of ammonia (which is calculated using turbulent diffusion and surface characteristics in models) might vary significantly depending on the season and region (Flechard et al., 2011; Aksoyoglu and Prévôt, 2018). Therefore, different regional parameters used in the calculations might lead to different model performance for deposition. Moreover, as reported by Theobald et al. (2019), the tendency of models to underestimate wet deposition and overestimate atmospheric concentrations is likely due to deficiencies in simulating wet-deposition processes, which are related to the vertical concentration profiles, scavenging coefficients or in-cloud processes, including the parameterization of clouds.

Evaluation of total nitrogen deposition is challenging because of a lack of measurements, especially of dry deposition, estimates are therefore based on the concentrations and deposition velocities (Flechard et al., 2011). On the other hand, model performance for wet deposition depends strongly on the performance of the meteorological model (Vivanco et al., 2018). Our model results for wet deposition of WNHx, WNOx and WSOx in 2010 are shown together with the available measurements in Fig. S1. The correlation between model results and measurements for wet deposition is high (between 0.61 for $WSO_x$ and 0.81 for $WNH_x$), the wet deposition of all three species, however, is underestimated with the mean fractional bias (MFB) ranging from -40% to -58% (Table S2). In the Eurodelta-Trends model-intercomparison study, a significant negative correlation was found for reduced nitrogen, i.e. the tendency that more the model underestimates the wet deposition, the more it overestimates the atmospheric concentrations (Theobald et al., 2019).

Measurements of total nitrate and ammonia concentrations are mainly available for northern Europe and they have very little overlap with the wet-deposition sites in central and western Europe (see Figs. 1b, c and Fig. S1). Comparison of model performance for TNHx and WNHx with observations at 15 common sites (Fig. S2) suggests that at 9 stations overestimation of TNHx might partially be attributed to underestimation of WNHx (with the largest anti-correlation of bias at sites in the Czech Republic and Poland), as also found by other models used in the EDT model intercomparison study (Theobald et al., 2019).

Among the SIA components, the best agreement between model and measurements is for sulfate (MFB = 4.7%) (Table S2, Fig. 1). The modeled concentrations of the other SIA components - for which the spatial coverage in central and western Europe is rather poor - are higher than the measured ones, especially for nitrate (MFB = 54.6%) (Fig. 1, Table S2). Overall, the performance of CAMx model in this study is similar to the other models participating in the EDT project (Ciarelli et al., 2019; Theobald et al., 2019).

## 3.2 Changes in concentrations

### 3.2.1 Gaseous species: $NH_3$ and $HNO_3$

The highest ammonia concentrations are predicted around the Benelux area and northern Italy where emissions are the highest in Europe (Fig. 2a, left panel). The model results suggest that between 1990 and 2000, the annual average concentrations of ammonia decreased only around the Netherlands and in the eastern part of the model domain by about 30-40%, while an increase was predicted in the rest of the domain (Fig. 2b, left panel, see Fig. S3a for relative changes). This is consistent with the observed trends in Europe during the same period (Colette et al., 2016). Ammonia concentrations decreased further around the Netherlands and started to decline also in northern Italy between 2000 and 2010, while they continued to increase in other parts of Europe (Fig. 2c, left panel). The predictions based on the current legislation (CLE) scenario emissions suggest that the changes during the period between 2010 and 2030 will be much smaller (Fig. 2d, left panel).

Concentrations of nitric acid ($HNO_3$) are generally higher in areas with both high $NO_x$ emissions and photochemical activity, such as coastal areas and the shipping routes especially in the Mediterranean region (Fig. 2a, right panel). The large decline (50-60%) in nitric acid concentrations in those areas between 1990 and 2000 is mainly due to a decrease in $NO_x$ emissions as a result of significant reductions in emission standards in Europe (Figs. 2b, right panel, S3b). The results show an increase in nitric acid concentrations later between 2000 and 2010 around the English Channel, North Sea, Baltic Sea and the Mediterranean Sea as well as northern Italy (Fig. 2c, right panel). These results suggest that decreased ammonia concentrations in the same period around the Netherlands, Denmark and northern Italy led to a lower ammonium nitrate formation and higher nitric acid concentrations in the air. Our calculations suggest that with further reductions in $NO_x$ emissions, nitric acid levels would continue to decrease until 2030 due to reductions in both land and ship emissions (Fig. 2d, right panel). On the other hand, since simulations for 2030 were performed using the meteorological parameters of 2010, one should keep in mind that potentially higher temperatures in the future might increase the evaporation of ammonium nitrate to form its gaseous components $NH_3$ and $HNO_3$ (Fowler et al., 2015).

### 3.2.2 Secondary inorganic aerosols

Annual average concentrations of SIA components (PNH4, PNO3, PSO4) in 1990, as well as changes in their concentrations between 1990-2000, 2000-2010 and 2010-2030_CLE are shown in Fig. 3. The highest ammonium

concentrations are predicted for central Europe and the Po Valley (Fig. 3a, left panel). The concentrations in eastern Europe are relatively high as well. As seen in the left panel of Fig. 3b, the ammonium concentrations decreased significantly (by 40-50%) between 1990 and 2000 in central and eastern Europe (see Fig. S3c for relative changes) while there was some increase in the east after 2000 (Fig. 3c, left panel). Simulations using the CLE emission scenarios suggest that the ammonium concentrations will continue to decrease between 2010 and 2030 with no more increase anywhere in the whole model domain (Fig. 3d, left panel). A similar trend can be observed in the particulate nitrate concentrations, with a significant decrease over the continent between 1990 and 2010 (Fig. 3 a-c, middle panels). A slight increase, however, is noticeable over the Baltic Sea and Scandinavia between 1990 and 2000 as well as in the eastern part of the model domain between 2000 and 2010. On the other hand, particulate nitrate concentrations will decrease in all of Europe between 2010 and 2030 (Fig. 3d, middle panel).

The concentrations of particulate sulfate ($PSO_4$) in 1990 were higher in central and eastern Europe, the Balkans and along shipping routes than the rest of Europe (Fig. 3a, right panel). They decreased continuously (by 60-70%) over the period between 1990-2010 (Fig. 3b-c, right panels, Fig. S3e). Results of future scenario simulations suggest that sulfate concentrations will continue to decrease in central Europe as well as along shipping routes until 2030 assuming a current legislation (CLE) scenario (Fig. 3d, right panel).

The results obtained from these simulations indicate that a significant reduction (> 40%) in the secondary inorganic aerosol concentrations was achieved especially between 1990 and 2000 (Fig. S3f), consistent with the larger reductions in emissions during the first decade (Table 2). Continuous reductions of land and ship emissions until 2030 will lead to a further decrease in SIA concentrations in Europe. The model results suggest that the relative composition of secondary inorganic aerosols will be different in 2030 compared to 1990 due to a larger decrease in sulfate concentrations than nitrate in most of the model domain. An example for the change in the SIA composition is shown for Payerne, a rural site in Switzerland, in Fig. S4. The sulfate fraction decreases from 30% to 21% while the particulate nitrate fraction increases from 47% to 56% between 1990 and 2030, assuming the CLE scenario.

### 3.3 Changes in deposition

### 3.3.1 Dry deposition

A large fraction of the total nitrogen deposition in Europe is due to the deposition of reduced nitrogen compounds, with dry deposition of ammonia being the dominant one (Aksoyoglu et al., 2014; Aksoyoglu and Prévôt, 2018; Simpson et al., 2014). Removal of ammonia from the atmosphere through dry deposition is quite fast - *i.e.* deposition occurs in areas close to the emission sources (Fig. 4a, left panel). The model results show that dry deposition of ammonia decreased around the Netherlands, northern Germany and slightly also in the eastern part of the model domain between 1990 and 2000 while there was an increase in the rest of Europe (Fig. 4b, left panel). After 2000, however, dry deposition started increasing also in eastern Europe (Fig. 4c, left panel). Simulations for 2030 calculated by changing only the emissions according to the CLE scenario suggest that there will be a small decrease in dry deposition only in the north around the Netherlands and northern Germany between 2010 and 2030 (Fig. 4d, left panel). A slight increase along the coastal areas in western and northern Europe is probably caused by the changes in ship emissions (see Section 3.4).

**3.3.2 Wet deposition**

The amount of precipitation which is generated by the meteorological models is crucial for simulating wet deposition in air quality models. The performance evaluation of the accumulated precipitation used in the Eurodelta-Trends exercise is discussed in detail in Theobald et al. (2019). The model biases are very small for accumulated annual precipitation for the meteorological model used in this study; there is an underestimation of 4%-8%. The wet deposition of ammonium in 1990 is shown in Fig. 4a, right panel. The modeled wet deposition decreased in eastern Europe between 1990 and 2000 while it slightly increased around the English Channel and North Sea (Fig. 4b, right panel). The decrease in the east is in line with the largest emission reductions in that area. The increase in wet deposition in the north around the English Channel could be due to increased precipitation between 1990-2000 (Theobald et al., 2019). Increased emissions from ships, however, could also be the reason for the increased wet deposition of ammonium. After 2000, there was a decrease in wet deposition in most of the domain (Fig. 4c, right panel). Other models which participated in the EDT project and simulated the whole 21-year period between 1990 and 2010, found similar results with a decrease in wet deposition in the east and an increase in north-west Europe during the first period between 1990-2000 (Theobald et al., 2019). Assuming the CLE scenario (using the meteorology of 2010), wet deposition of ammonium is predicted to decrease in Europe significantly (20-40%) between 2010 and 2030 (Fig. 4d, right panel). The change in wet deposition between 1990 and 2010 might be due to a change in both the air concentrations and the amount of precipitation, but it can only be due to a change in the concentrations between 2010 and 2030_CLE since the same meteorological parameters were used for both years.

**3.4 Effects of ship emissions**

The two scenarios 2020_CLE and 2030_CLE take both land and ship emissions into account according to the current legislation. In order to investigate the effect of ship emissions on the gaseous and particulate species, we compared two ship emission scenarios for both 2020 and 2030 as described in Table 1. In scenarios 2020_CLEland and 2030_CLEland, ship emissions were kept the same as in 2010 while they were projected to 2020 and 2030 using the CLE scenario in 2020_CLE and 2030_CLE, respectively. The difference in concentrations between CLE and CLEland scenarios therefore shows the effect due to the changes in ship emissions in the corresponding years (Fig. 5, Fig. S5).

In all three cases (2020, 2030_CLE and 2030_MTFR), gaseous ammonia concentrations are predicted to increase due to changes in ship emissions especially around the Benelux area and along the Mediterranean coast (Fig. 5a). On the other hand, nitric acid concentrations will decrease in the North Sea, Baltic Sea and the Mediterranean Sea, and increase along the Atlantic coast under the assumptions of the CLE scenario in 2020 and 2030 (Fig. 5b). The effect of emissions from international shipping activities along the Atlantic coast and Gibraltar Strait can also be seen in the particulate nitrate concentrations (Fig. S5d). The importance of shipping activities due to their relatively high $NO_x$ emissions was also reported for south-west Europe by Pay et al. (2019). Our scenario calculations suggest that when ship emissions are reduced according to the MTFR scenario in 2030, nitric acid and particulate nitrate would no longer increase along the Atlantic coast but decrease (see Fig. 5b, Fig. S5d).

The decrease in concentrations of secondary inorganic aerosols along the coastal areas -especially with the MTFR scenario- (Figs. 5c, S5) is due to a significant decrease in nitrate and sulfate concentrations around the Benelux region and the Mediterranean, respectively, as shown in Jiang et al. (2020). These results suggest that when ship

emissions are reduced further in the future, particulate ammonium formation will decrease especially around the

Benelux area, North Sea, Baltic Sea, English Channel and the Mediterranean region, leaving more ammonia in the gas phase (Figs. 5a, S5a). This would then lead to an increase in the dry deposition of ammonia along the coastal areas (Fig. 5d, S5g). On the other hand, wet deposition of ammonium will decrease along the Scandinavian and Adriatic coasts due to future reductions of ship emissions (Fig. S6).

**3.5 Sensitivity of aerosol formation to $NO_x$ and $NH_3$ emissions**

SIA formation depends strongly on its precursor emissions. Ammonia reacts rapidly with atmospheric sulfuric and nitric acids to form ammonium sulfate and ammonium nitrate (Behera et al., 2013). Reaction with sulfuric acid (or with ammonium bisulfate) is favored over the reaction with nitric acid; ammonium nitrate is formed only after all sulfate is neutralized by $NH_3$. Ammonium nitrate formation, which is favored by low temperatures and high relative humidity is in a reversible equilibrium with ammonia and nitric acid.

An earlier model study covering Switzerland and northern Italy, using emissions from the year 2000, showed that aerosol formation was rather limited by $NO_x$ emissions in northern Switzerland while it was dependent on both $NO_x$ and $NH_3$ emissions in northern Italy (Andreani-Aksoyoglu et al., 2008). Other sensitivity studies performed over the whole European domain for the period 2004-2006 suggested that SIA formation was more sensitive to $NH_3$ emissions in most of Europe, except the Netherlands, northern Switzerland and north-western France where

ammonia emissions are high (Aksoyoglu et al., 2011; Pay et al., 2012). Since sulfate and nitrate aerosol formation depends on the availability of their precursors $SO_2$, $NO_x$ and $NH_3$, changes in emissions since the 1990s might have affected the sensitivity of aerosol formation in Europe.

We tested the sensitivity of aerosol formation to emissions by reducing $NO_x$ and $NH_3$ emissions by 30% in two separate simulations for the past (1990) and the future (2030_CLE). The change in the annual average SIA

concentrations for these simulations is shown in Fig. 6. Decreasing $NH_3$ emissions are predicted to be more effective in reducing SIA concentrations in 1990 for a large part of the model domain (Fig. 6, upper panels) as also reported in Aksoyoglu et al. (2011). In 2030_CLE, however, the effectiveness of emission reductions looks different as a result of the larger reductions in $NO_x$ emissions compared to $NH_3$ emissions in Europe (Fig. 6, lower panels). The change in the colors from red in 1990 to blue in 2030_CLE in central and western Europe as well as

the UK (right panels) suggests that aerosol formation will become more $NO_x$-sensitive in 2030 in those areas while in the eastern part of the model domain it will still be more sensitive to ammonia emissions (Fig. 6, lower right panel). It should be noted however, that the sensitivity to emissions is weaker in 2030_CLE than in 1990 (see scales in Fig. 6).

**4 Conclusions**

In this study we investigated the role of ammonia in European air quality by means of CAMx model simulations for the period between 1990 and 2030 with 10-year intervals. We analyzed the modeled annual average concentrations of ammonia ($NH_3$), nitric acid ($HNO_3$), secondary inorganic aerosol (SIA) and its components as well as dry ammonia and wet ammonium deposition. The model results suggest that the decrease in SIA concentrations by about 40% between 1990 and 2010 was mainly due to reductions of $NO_x$ and $SO_2$ emissions in

Europe. Ammonia concentrations, on the other hand, decreased by about 30% only around high emission areas

(Benelux area, northern Italy) while there was a slight increase in the other parts of the model domain, leading also to an increase in dry deposition of ammonia. The modeled changes in the annual concentrations were larger for the first decade (1990-2000), especially for nitric acid, particulate ammonium and sulfate. On the other hand, changes during the period between 2010 and 2030, assuming the current legislation (CLE) scenario, are predicted

to be smaller than those achieved earlier for all analyzed species in this study.

Simulations using the current legislation (CLE) and maximum technically feasible reduction (MTFR) scenarios for 2020 and 2030 suggest that when ship emissions will be regulated more strictly in the future, particle formation will decrease, especially around the Benelux area, North Sea, Baltic Sea, English Channel and the Mediterranean region, leaving more ammonia in the gas phase, which will lead to an increase in dry deposition. In the north, the

decrease in aerosol concentrations will be mainly due to reduced particulate nitrate formation, while the change in the Mediterranean area will be caused mainly by decreased sulfate aerosols.

In order to investigate whether the sensitivity of aerosol formation to $NO_x$ or $NH_3$ emissions changes during the period between 1990 and 2030, we performed sensitivity tests by repeating the simulations for 1990 and 2030_CLE with $NO_x$ and $NH_3$ emissions reduced by 30%, separately. In 1990, SIA formation was more affected

by $NH_3$ emission reductions in central Europe because of relatively high $NO_x$ emissions at that time. In 2030, however, $NO_x$ emission reductions reduce SIA concentrations more than $NH_3$ reductions. These results indicate a shift in the sensitivity of aerosol formation from $NH_3$ towards $NO_x$ emissions in a large part of Europe between 1990 to 2030 due to a larger change in $NO_x$ levels during that period compared to changes in $NH_3$ concentrations.

*Data availability:* Data will be available online before publication on ACP

*Supplement link:*

*Author contribution:* SA and JJ developed the idea and analyzed the results, JJ performed the simulations, SA wrote the paper, GC contributed to input preparation, SA and ASHP supervised the project. All authors contributed to the text, interpretation of the results and review of the article.

*Competing interests:* The authors declare that they have no conflict of interest

*Acknowledgements:* We acknowledge the EURODELTA-Trends project for providing meteorological data, anthropogenic emissions and boundary conditions as model input for 1990–2010, the National Aeronautics and Space Administration (NASA) and its data-contributing agencies (NCAR, UCAR) for the TOMS and MODIS data and the TUV model, the International Institute for Applied Systems Analysis (IIASA) for the GAINS ship

emissions in 2020 and 2030. We would like to thank RAMBOLL for the continuous support of the CAMx model. Model simulations were performed at the Swiss National Supercomputing Centre (CSCS). This study was financially supported by the Swiss Federal Office for the Environment (FOEN). Giancarlo Ciarelli acknowledges the support of the Swiss National Science Foundation (grant no. P2EZP2_175166).

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

**Table 1.** Description of simulations. CLE: Current Legislation, MTFR: Maximum Technically Feasible Reductions

| Name | Meteorology | Boundary conditions | Anthropogenic Emissions |
|---|---|---|---|
| **Base Cases** | | | |
| 1990 | 1990 | 1990 | 1990 |
| 2000 | 2000 | 2000 | 2000 |
| 2010 | 2010 | 2010 | 2010 |
| **Scenarios** | | | |
| 2020_CLE | 2010 | 2010 | 2020CLE |
| 2020_CLEland | 2010 | 2010 | 2020CLE land, 2010 ship emissions |
| 2030_CLE | 2010 | 2010 | 2030CLE |
| 2030_CLEland | 2010 | 2010 | 2030CLE land, 2010 ship emissions |
| 2030_MTFR | 2010 | 2010 | 2030CLE land, MTFR ship emissions |
| **Sensitivity Tests** | | | |
| 1990_70%$NO_x$ | 1990 | 1990 | 1990, $NO_x$ emissions reduced by 30% |
| 1990_70%$NH_3$ | 1990 | 1990 | 1990, $NH_3$ emissions reduced by 30% |
| 2030_70%$NO_x$ | 2010 | 2010 | 2030CLE, $NO_x$ emissions reduced by 30% |
| 2030_70%$NH_3$ | 2010 | 2010 | 2030CLE, $NH_3$ emissions reduced by 30% |

**Table 2.** Differences in emissions between various years and future scenarios

| | Changes in emissions (%) | | |
|---|---|---|---|
| | $NO_x$ | $SO_x$ | $NH_3$ |
| 1990-2000 | -23.7 | -51.0 | -18.6 |
| 2000-2010 | -16.8 | -35.3 | -5.3 |
| 1990-2010 | -36.5 | -68.3 | -23.0 |
| 2010-2020_CLE | -28.1 | -41.5 | -4.1 |
| 2010-2030_CLE | -39.7 | -47.5 | -10.8 |
| 2010-2030_MTFR | -52.5 | -49.7 | -10.8 |






## Figure captions:

**Figure 1: Mean bias between measured and modelled concentrations of (a) ammonia (ppb), (b) total ammonia ($\mu$g N m$^{-3}$), (c) total nitrate ($\mu$g N m$^{-3}$), (d) ammonium ($\mu$g m$^{-3}$), (e) particulate nitrate ($\mu$g m$^{-3}$) and (f) sulfate ($\mu$g m$^{-3}$) at various sites in 2010.**

**Figure 2: Ammonia (left) and nitric acid (right): (a) annual average concentrations (ppb) in 1990, changes (in ppb) between (b) 1990-2000, (c) 2000-2010 and (d) 2010-2030_CLE.**

**Figure 3: Ammonium (left), particulate nitrate (middle) and sulfate (right): (a) annual average concentrations ($\mu$g m$^{-3}$) in 1990, changes ($\mu$g m$^{-3}$) between (b) 1990-2000, (c) 2000-2010 and (d) 2010-2030_CLE.**

**Figure 4: Dry deposition of ammonia (left panels) and wet deposition of ammonium (right panels): (a) accumulated deposition (kg N ha$^{-1}$) in 1990, changes (in kg N ha$^{-1}$) between (b) 1990 and 2000, (c) 2000 and 2010 and (d) 2010 and 2030_CLE.**

**Figure 5: Effect of ship emissions on annual average concentrations of (a) ammonia (ppb), (b) nitric acid (ppb), (c) secondary inorganic aerosols ($\mu$g m$^{-3}$) and (d) on dry deposition of ammonia (kg N ha$^{-1}$) in scenarios 2020 CLE (left), 2030 CLE (middle) and 2030 MTFR (right). CLEland: with ship emissions of 2010; CLE: with ship emissions under current legislation; MTFR: with ship emissions under maximum technically feasible reductions.**

**Figure 6: Change in SIA concentrations ($\mu$g m$^{-3}$) due to 30% reduction of NO$_x$ emissions (left), due to 30% reduction of NH$_3$ emissions (middle) and the difference between the two figures (right) (a) in 1990 and (b) in 2030_CLE. Red color in the right panels shows NH$_3$-sensitive areas; blue color shows NO$_x$-sensitive areas.**

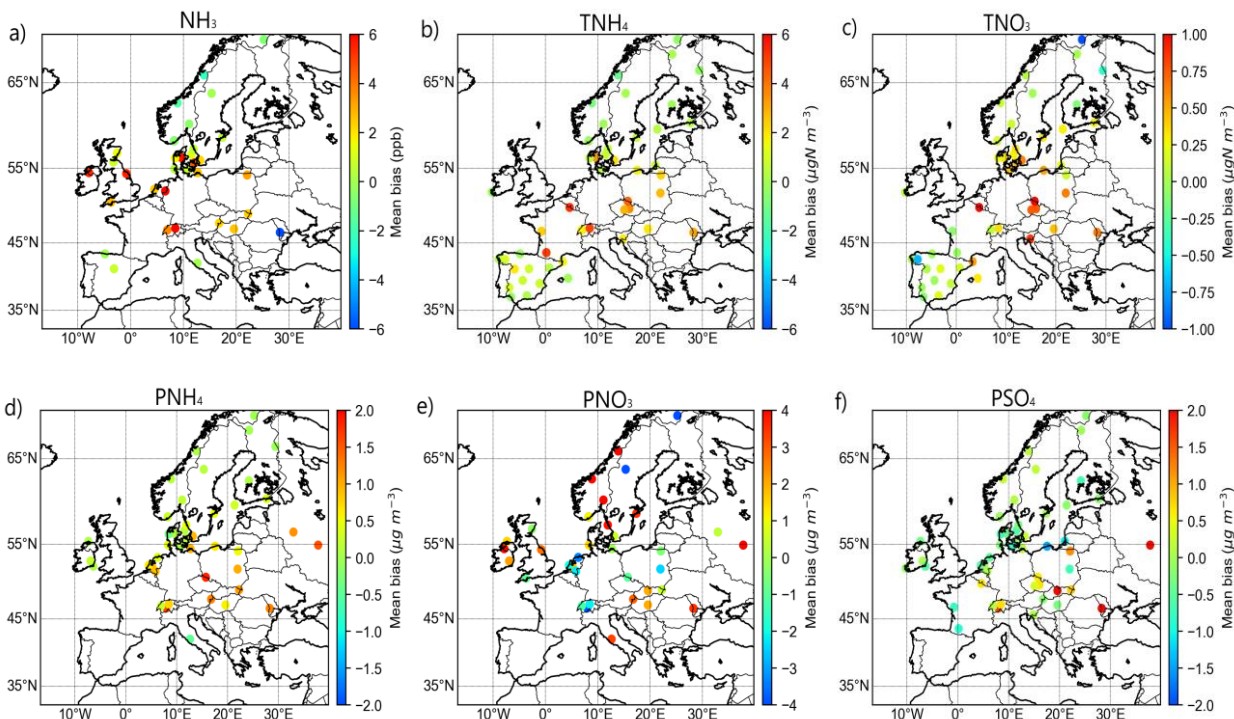

**Figure 1: Mean bias between measured and modelled concentrations of (a) ammonia (ppb), (b) total ammonia (µg N m⁻³), (c) total nitrate (µg N m⁻³), (d) ammonium (µg m⁻³), (e) particulate nitrate (µg m⁻³) and (f) sulfate (µg m⁻³) at various sites in 2010.**



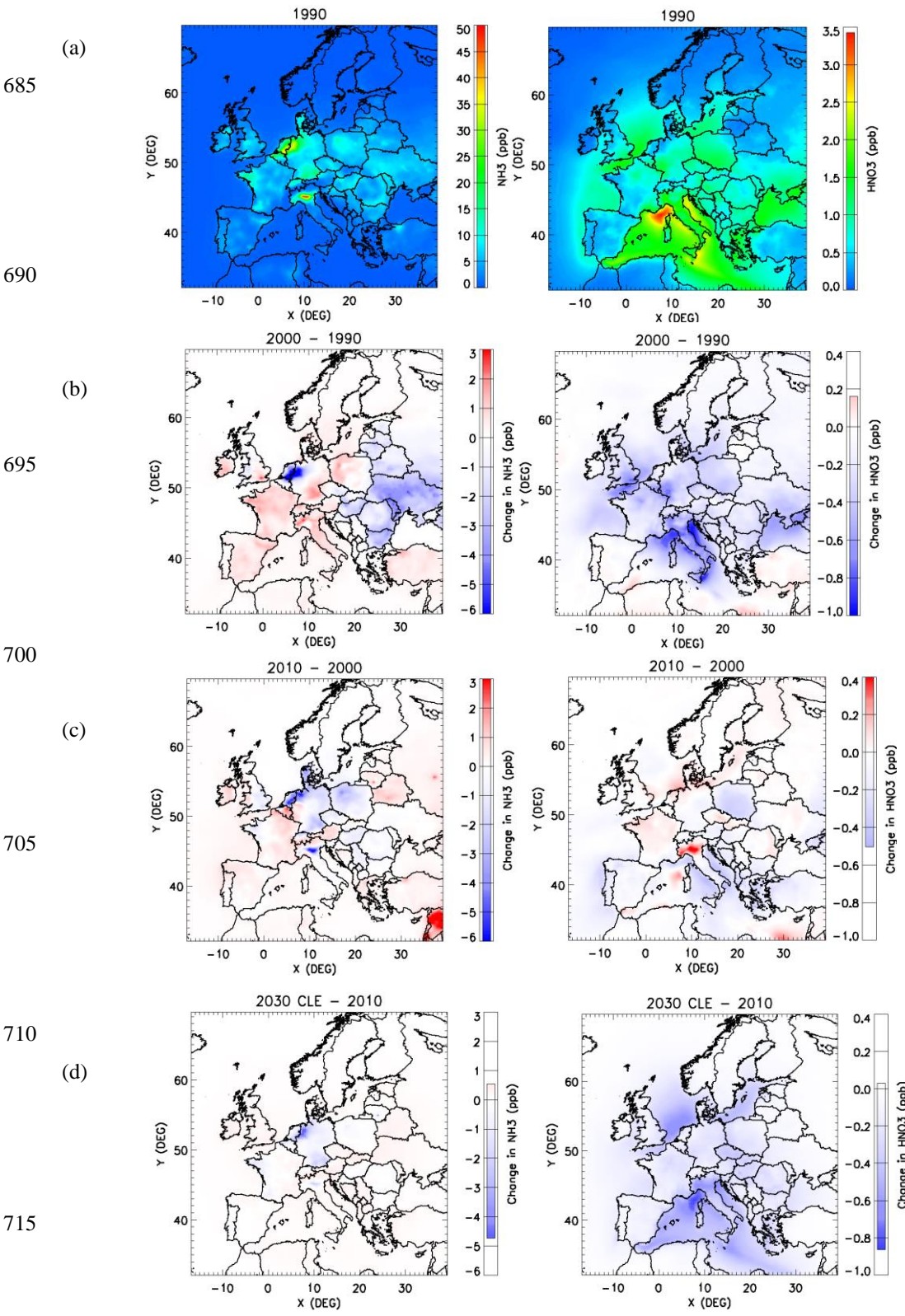








**Figure 2: Ammonia (left) and nitric acid (right): (a) annual average concentrations (ppb) in 1990, changes (in ppb) between (b) 1990-2000, (c) 2000-2010 and (d) 2010-2030_CLE.**


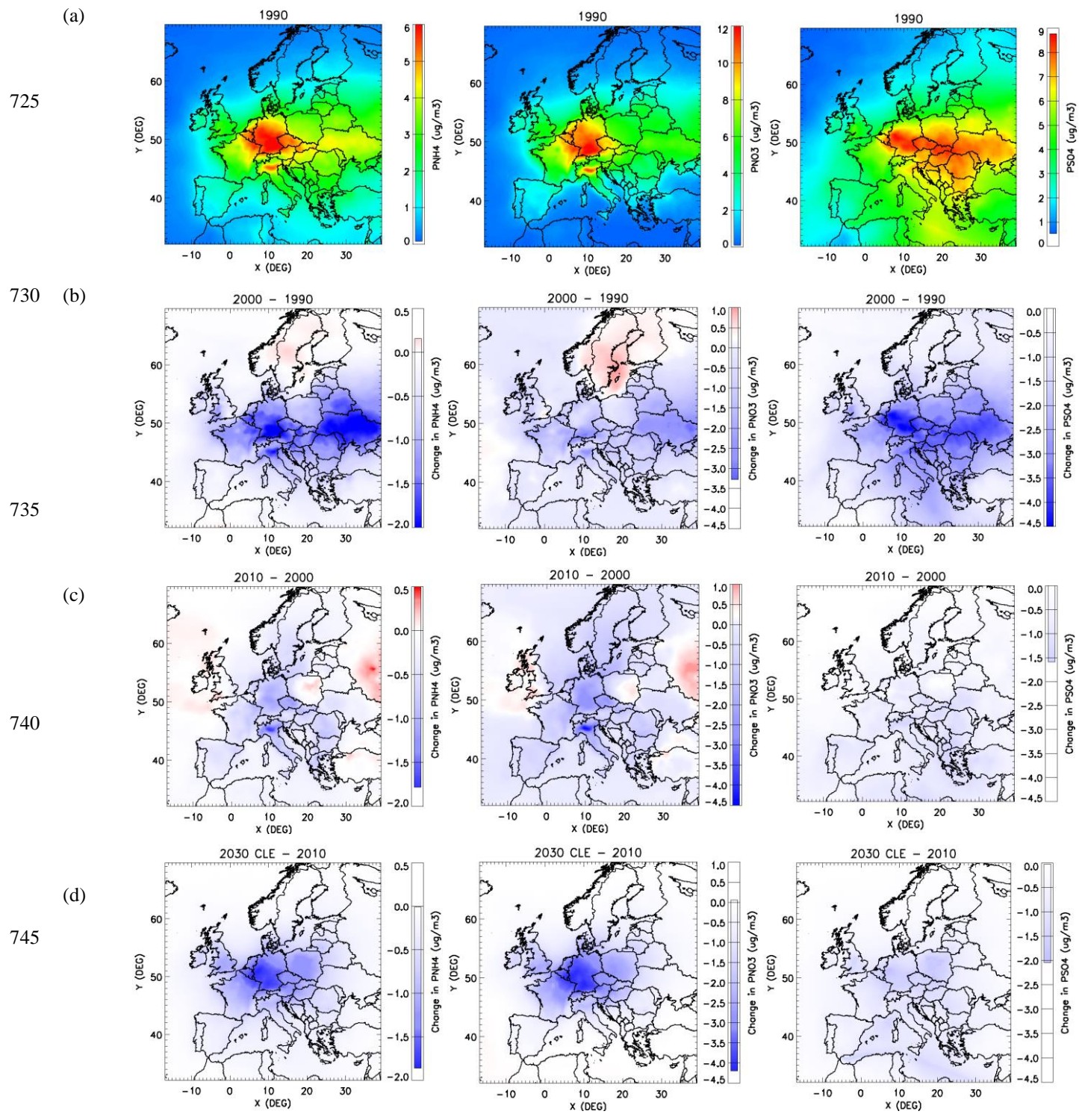

Figure 3: Ammonium (left), particulate nitrate (middle) and sulfate (right): (a) annual average concentrations (μg m$^{-3}$) in 1990, changes (μg m$^{-3}$) between (b) 1990-2000, (c) 2000-2010 and (d) 2010-2030_CLE.

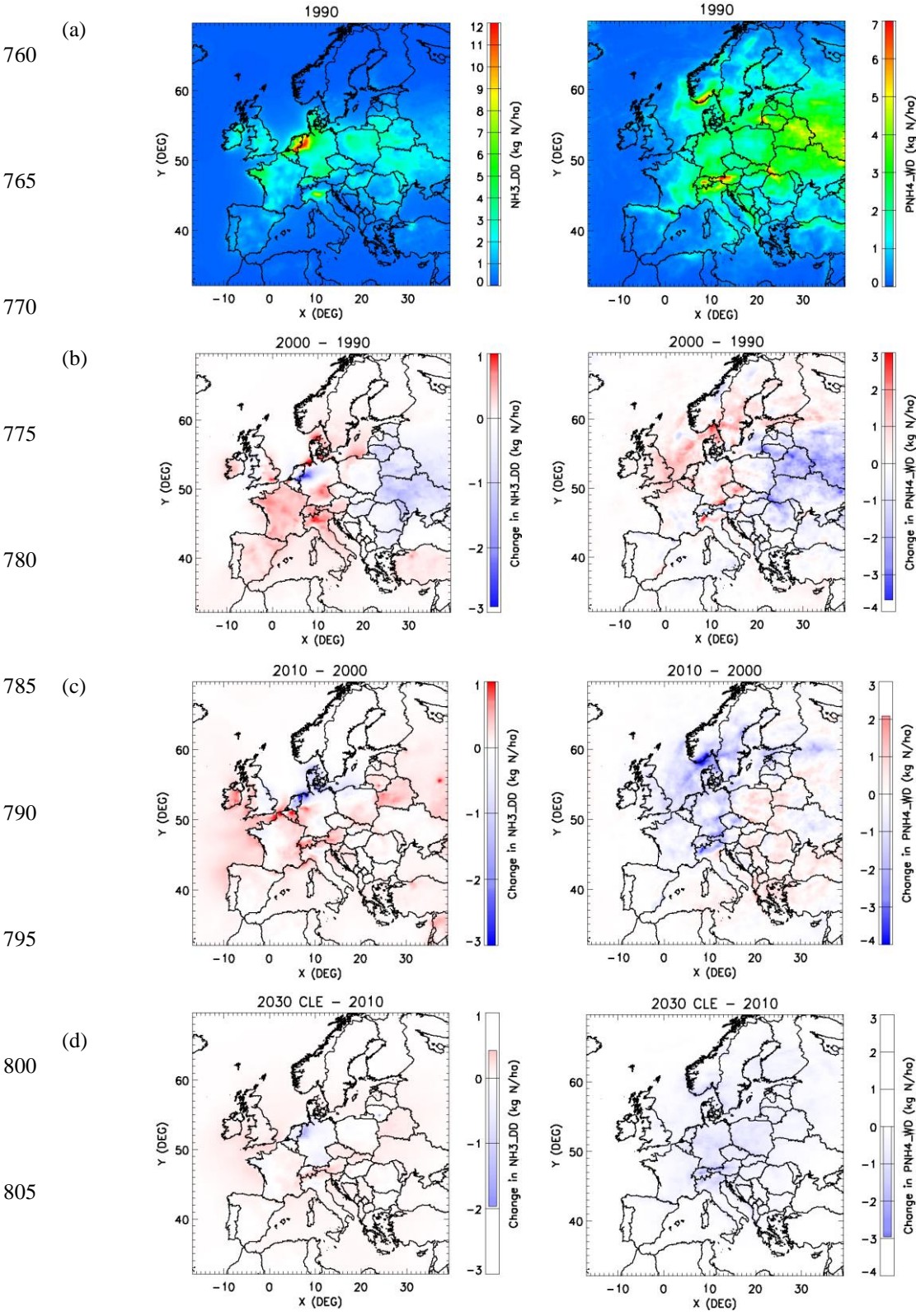

**Figure 4: Dry deposition of ammonia (left panels) and wet deposition of ammonium (right panels): (a) accumulated deposition (kg N ha[-1]) in 1990, changes (in kg N ha[-1]) between (b) 1990 and 2000, (c) 2000 and 2010 and (d) 2010 and 2030_CLE.**

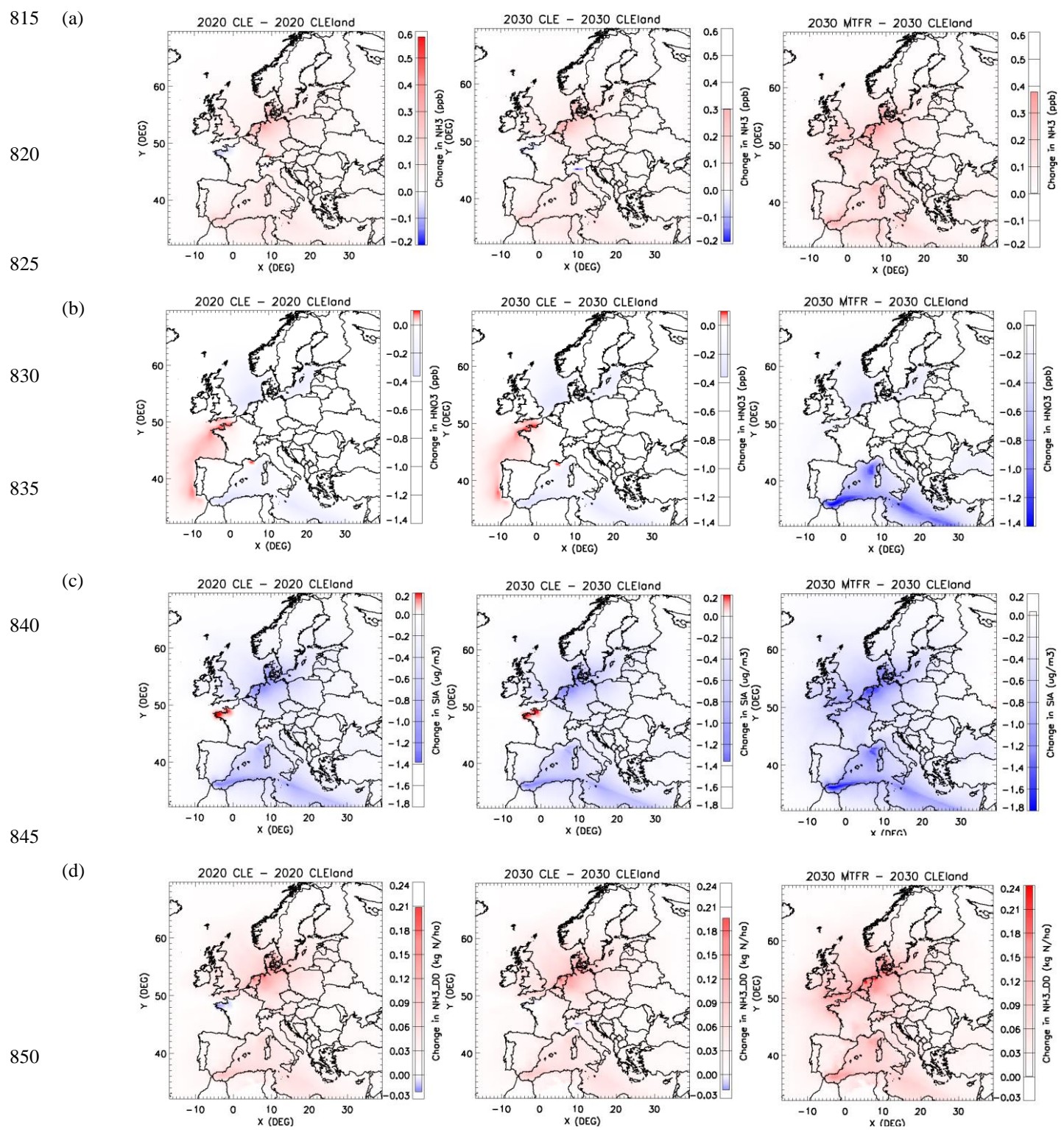

**Figure 5: Effect of ship emissions on annual average concentrations of (a) ammonia (ppb), (b) nitric acid (ppb), (c) secondary inorganic aerosols (µg m⁻³) and (d) on dry deposition of ammonia (kg N ha⁻¹) in scenarios 2020 CLE (left), 2030 CLE (middle) and 2030 MTFR (right). CLEland: with ship emissions of 2010; CLE: with ship emissions under current legislation; MTFR: with ship emissions under maximum technically feasible reductions.**

(a)

(b)

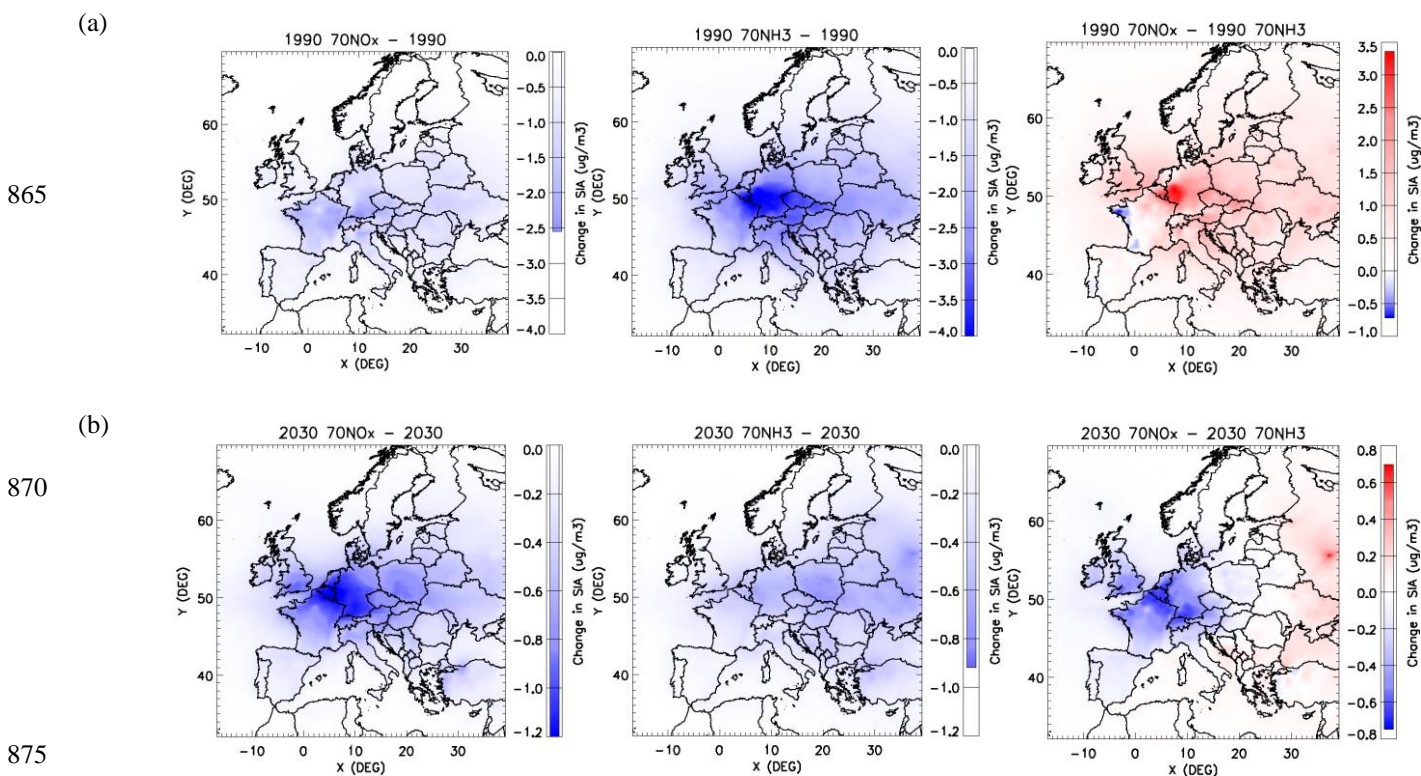

**Figure 6: Change in SIA concentrations (μg m⁻³) due to 30% reduction of NOₓ emissions (left), due to 30% reduction of NH₃ emissions (middle) and the difference between the two figures (right) (a) in 1990 and (b) in 2030_CLE. Red color in the right panels shows NH₃-sensitive areas; blue color shows NOₓ-sensitive areas.**