# Peer review of "Role of ammonia in European air quality with changing land and ship emissions between 1990 and 2030"

_Atmospheric Chemistry and Physics, 2020_

## Referee Comment (RC1) · Anonymous Referee #1 · 25 Sep 2020

Dear Editor,

this MS presents a modelling study dealing with the role of ammonia on air quality in Europe, with a focus on shipping as a key emission sector. The text is straightforward and well-written, and of interest to the scientific community. I have only minor issues which should be clarified prior to publication:

- line 51: what is the reason behind the increase in ammonia emissions since 2014?

- line 67: what is the status of this implementation? Have these new sulfur emission regulations been effectively implemented (as they were supposed to start in 2020)?

[Figure]

- line 77: same as above, what is the status of this statement? "According to the European Environment Agency, emissions of nitrogen oxides from international maritime transport in European waters are projected to increase and could be equal to land-based sources by 2020 (EEA, 2013). " The reference dates back to 2013, do the authors have data on the current emissions? How accurate was EEA's projection from 2013?

- line 120: the scenarios "current legislation (CLE)" refer to the regulations included the lower sulfur limits from 2020 (see comment to line 67)? Or prior to 2020? Please clarify.

- line 135, please review sentence (2 verbs): "The model results for 1990, 2000 and 2010 were compared with the measurements available at the EDT project database based on EMEP datasets and model performance for SO2, NO2, PM10, PM2.5 and hourly O3 was discussed in detail in Jiang et al. (2020)."

- line 140: measuring ammonia is rather complex, therefore the quality of the observations should also be discussed (even if briefly). Please add some information on the measurement method and comparability between (the few) ammonia datasets available.

- line 156: "and/or deposition is underestimated by the model for which the resolution might also be critical factor". Is the model resolution not the same for all regions? If it is, then it could not explain the differences between central Europe and Iberian Peninsula and in Scandinavia (Fig. 1). Can the authors provide an explanation about why deposition might be more underestimated in central Europe than in the Iberian Peninsula and in Scandinavia? This seems a more likely cause for the model's overestimation around these high emission areas.

- line 171: "Among the SIA components, the best agreement between model and measurements is for sulfate". Can the authors quantify the relative difference (in %) between modelled and measured concentrations, for sulfate and nitrate respectively? It

would be useful for the reader to have this information as well as the absolute difference (in microg/m3 shown in the Figure). Also for ammonia, in the previous paragraphs.

- lines 180-185: similarly to above, what is the reason for the increases in ammonia emissions? Is it increases in key sources (agriculture)? Or to a mix of emissions and atmospheric processes?

- line 197: "On the other hand, since simulations for 2030 were performed using the meteorological parameters of 2010, one should keep in mind that potentially higher temperatures in the future might increase the evaporation of ammonium nitrate to form its gaseous components NH3 and HNO3". This is a key point which could be highlighted in the abstract.

- line 217: as above, please clarify what is meant by "current legislation" (before or after 2020): "Results of future scenario simulations suggest that sulfate concentrations will continue to decrease in central Europe as well as along shipping routes until 2030 assuming a current legislation (CLE) scenario (Fig. 3d, right panel)"

- section 3.4: only as a suggestion, it might have been interesting to add an additional scenario including the implementation of a SECA in the Mediterranean (Rouïl, L., Ratsivalaka, C., André, J.-M., Allemand, N., 2019. ECAMED: a Technical Feasibility Study for the Implementation of an Emission Control Area (ECA) in the Mediterranean Sea. IMO report MEPC 74/INF.5.). An analysis of the potential impacts/benefits of this potential SECA in the framework of the authors' study could be very useful.

- Table 1: please define the acronyms (CLE, MTFR) in the table header.

---

## Referee Comment (RC2) · Anonymous Referee #2 · 6 Oct 2020

This paper is a nice overview of the changing role of ammonia in the European atmosphere. I think it is generally well written, and my only major criticism is that a number of important uncertainties which might affect the conclusions are not discussed or addressed.

**Major comments**

The major areas of uncertainty that I missed include:

1. Bi-directionality. It is well established that ammonia can be emitted as well as deposited as a result of the equilibria between atmospheric and surface concentrations. It displays a so-called compensation point (Nemitz et al., 2001; Flechard et al., 1999, 2013), and this can affect the deposition close to source areas and long-range transport in general (Bash et al., 2013; Wichink Kruit et al., 2012).

2. This study seems to ignore the impacts of co-deposition, in which the acidity of the surface (affected by both $SO_2$ and $NH_3$ emissions, and their trends) changes. The impacts of this process on trends have been explored in for example Wichink Kruit et al. (2017).

3. A similar issue with trends, also not mentioned, is changing pH of rainwater (Banzhaf et al., 2012).

4. Meteorological variability. The current study mainly uses meteorology from just two years, 1990 and 2010, but Wichink Kruit et al. (2017) showed that meteorology can also account for a significant contribution to $NH_3$ trends.

5. Ship-plumes. It is well known that models tend to mis-represent $HNO_3$ production from NO emitted from ships into clean marine environments (von Glasow et al., 2003; Vinken et al., 2011, 2014). This could potentially have been handled with the CAMx model's plume-in-grid approach, but this doesn't seem to have been used. However, some of the comments made about $HNO_3$ (e.g. L187 onwards) may be impacted by this issue.

6. In the introduction, I missed some discussion of trend studies on land-based emissions and concentration/deposition trends which have already been done, e.g. Fowler et al. (2007); Fagerli and Aas (2008) or Wichink Kruit et al. (2017). How does the current study add to these? (Page 3 gives a lot of information given on the impacts of shipping, but not much about land.)

**Smaller comments**

1. L36. The Maas & Grennfelt reference is not peer-reviewed. There are plenty of peer-reviewed publications on this subject.

2. L37. The authors only mention ammonium sulfate here, but bi-sulfate is an important component of European aerosol too.

3. L42. The Dentener ref is 14 years old now; find something more

4. L53. Are you sure it is ammonium sulfate, and not bisulfate?

5. L54. The Colette et al 2016 reference seems to be some grey literature, with no address and no url. What is this? And surely there are some peer-reviewed papers that be cited to support this statement?

6. L65-17. The SECA's came into effect at the start of 2015

7. L80-84. It is unclear where the cited 1-14% PM2.5 applies. This number sounds very different to those cited for Karl et al., and so this paragraph is a little confusing. Are the Karl et al results similar to, or very different from those cited from Viana et al.?

8. L100-102 How is the coarse-mode aerosol (e.g. for nitrate) handled in this model system?

9. L103. Be explicit with a reference to the Zhang scheme (not just the cited CAMx user's guide). And whether co-deposition is included or not?

10. L134 on. Brief details on the measurement networks underlying EDT work should be given.

11. L140 I won't repeat Ref #1's comments, but agree with them.

12. L145. As noted above, many processes not discussed in this manuscript might also contribute to model-measurement bias. Another issues is scale, which is very briefly mentioned on L156, but which can be a very important factor for $NH_3$ (Theobald et al., 2016; Denby et al., 2020).

13. Notation. Better to use pNH4, pNO3, pSO4 than PNH4 etc, to avoid mixing chemical and atmospheric nomenclature.

14. L242 states that the amount of precipitation is crucial, but no figures are given on this here; please expand.

15. Many of the figure legends and colors need to be re-done. For example, in Fig2c reds are used for positive values and blues for negative, which is great, but in Figs. 2d and 2e the color-scale shows white for levels both above and below certain thresholds! Later Figures also show such strange behaviors. I suggest using the same color-scale for all subplots, and do not have the same color for different values.

16. Fig. S2 - which measurements? Be explicit.

**References**

Banzhaf, S., Schaap, M., Kerschbaumer, A., Reimer, E., Stern, R., van der Swaluw, E., and Builtjes, P.: Implementation and evaluation of pH-dependent cloud chemistry and wet deposition in the chemical transport model REM-Calgrid, Atmos. Environ., 49, 378–390, https://doi.org/10.1016/j.atmosenv.2011.10.069, 2012.

Bash, J. O., Cooter, E. J., Dennis, R. L., Walker, J. T., and Pleim, J. E.: Evaluation of a regional air-quality model with bidirectional $NH_3$ exchange coupled to an agroecosystem model, Biogeosciences, 10, 1635–1645, https://doi.org/10.5194/bg-10-1635-2013, `http://www.biogeosciences.net/10/1635/2013/`, 2013.

Denby, B., Gauss, M., Wind, P., Mu, Q., Grøtting Wærsted, E., Fagerli, H., Valdebenito, A., and Klein, H.: Description of the uEMEP_v5 downscaling approach for the EMEP MSC-W chemistry transport model, Geoscientific Model Development Discussions, 2020, 1–38, https://doi.org/10.5194/gmd-2020-119, `https://gmd.copernicus.org/preprints/gmd-2020-119/`, 2020.

Fagerli, H. and Aas, W.: Trends of nitrogen in air and precipitation: Model results and observations at EMEP sites in Europe, 1980-2003, Environ. Poll., 154, 448–461, 2008.

Flechard, C. R., Fowler, D., Sutton, M. A., and Cape, J. N.: A dynamic chemical model of bi-directional ammonia exchange between semi-natural vegetation and the atmosphere, Q. J. R. Meteorol. Soc., 125, 2611–2641, 1999.

Flechard, C. R., Massad, R.-S., Loubet, B., Personne, E., Simpson, D., Bash, J. O., Cooter, E. J., Nemitz, E., and Sutton, M. A.: Advances in understanding, models and parameterizations of biosphere-atmosphere ammonia exchange, Biogeosciences, 10, 5183–5225, https://doi.org/10.5194/bg-10-5183-2013, `http://www.biogeosciences.net/10/5183/2013/`, 2013.

Fowler, D., Smith, R., Muller, J., Cape, J., Sutton, M., Erisman, J., and Fagerli, H.: Long Term Trends in Sulphur and Nitrogen Deposition in Europe and the Cause of Non-linearities, Water, Air, & Soil Pollution: Focus, 7, 41–47, `http://dx.doi.org/10.1007/s11267-006-9102-x`, 2007.

Nemitz, E., Milford, C., and Sutton, M. A.: A two-level canopy compensation point model for describing bi-directional biosphere-atmosphere exchange of ammonia, Q. J. R. Meteorol. Soc., 127, 815–833, 2001.

Theobald, M. R., Simpson, D., and Vieno, M.: Improving the spatial resolution of air-quality modelling at a European scale – development and evaluation of the Air Quality Re-gridder Model (AQR v1.1), Geoscientific Model Development, 9, 4475–4489, https://doi.org/10.5194/gmd-9-4475-2016, `http://www.geosci-model-dev.net/9/4475/2016/`, 2016.

Vinken, G. C. M., Boersma, K. F., Jacob, D. J., and Meijer, E. W.: Accounting for non-linear chemistry of ship plumes in the GEOS-Chem global chemistry transport model, Atmospheric Chemistry and Physics, 11, 11 707–11 722, https://doi.org/10.5194/acp-11-11707-2011, `http://www.atmos-chem-phys.net/11/11707/2011/`, 2011.

Vinken, G. C. M., Boersma, K. F., van Donkelaar, A., and Zhang, L.: Constraints on ship $NO_x$ emissions in Europe using GEOS-Chem and OMI satellite $NO_2$ observations, Atmospheric Chemistry and Physics, 14, 1353–1369, https://doi.org/10.5194/acp-14-1353-2014, `http://www.atmos-chem-phys.net/14/1353/2014/`, 2014.

von Glasow, R., Lawrence, M. G., Sander, R., and Crutzen, P. J.: Modeling the chemical effects of ship exhaust in the cloud-free marine boundary layer, Atmos. Chem. Physics, 3, 233–250, `http://www.atmos-chem-phys.net/3/233/2003/`, 2003.

Wichink Kruit, R. J., Schaap, M., Sauter, F. J., van Zanten, M. C., and van Pul, W. A. J.: Modeling the distribution of ammonia across Europe including bi-directional surface-atmosphere exchange, Biogeosciences, 9, 5261–5277, https://doi.org/10.5194/bg-9-5261-2012, 2012.

Wichink Kruit, R. J., Aben, J., de Vries, W., Sauter, F., van der Swaluw, E., van Zanten, M. C., and van Pul, W. A. J.: Modelling trends in ammonia in the Netherlands over the 1990-2014, Atmos. Environ., 154, 20–30, https://doi.org/10.1016/j.atmosenv.2017.01.031, 2017.

---

## Author Comment (AC1) · 4 Nov 2020

**Replies to the Anonymous Referee 1**

We thank the referee for the valuable comments which helped us to improve the manuscript. Please find below our responses (in black) after the referee comments (in blue). Changes in the revised manuscript are written in *italics.*

This MS presents a modelling study dealing with the role of ammonia on air quality in Europe, with a focus on shipping as a key emission sector. The text is straight forward and well-written, and of interest to the scientific community. I have only minor issues which should be clarified prior to publication:

- line 51: what is the reason behind the increase in ammonia emissions since 2014?
Agriculture is the main source of ammonia and emissions mainly result from the stabling of animals and the storage and application of animal manure. The application of inorganic N-fertilizers is also a source of ammonia emissions. Emissions decreased between 1990 and 2000 in Europe mainly due to declining numbers of animals. After 2000, the decrease in European countries slowed and emissions even started to increase, especially in the eastern part of Europe. The increase in ammonia emissions since 2014 is due to the difficulty in implementing further emission reductions, especially in the agriculture sector. We added the following sentence in the revised manuscript:
P3, L56
*Ammonia emissions have been increasing again since 2014, however, posing problems for Europe (NEC, 2019). This is mainly due to the difficulty in implementing additional emission reductions in the agriculture sector, especially in the housing of animals and the storage and application of animal manures.*

- line 67: what is the status of this implementation? Have these new sulfur emission regulations been effectively implemented (as they were supposed to start in 2020)?

The new regulation has been in force since 1 January 2020.  It reduces the limit for sulfur in fuel oil used in ships operating outside designated emission control areas to 0.50%. IMO reports that it worked with the Member States as well as the shipping and refining industries to identify and mitigate transitional issues so that ships meet the new requirements
https://www.imo.org/en/MediaCentre/PressBriefings/Pages/34-IMO-2020-sulphur-limit-.aspx). There are also guidelines being developed by IMO for consistent implementation of the MARPOL regulation coming into effect from 1 January 2020.  Monitoring, compliance and enforcement of the new limit is the responsibility of governments and national authorities of Member States that are parties to MARPOL Annex VI. Flag States (the State of registry of a ship) and Port States also have rights and responsibilities to enforce compliance.

- line 77: same as above, what is the status of this statement? "According to the European Environment Agency, emissions of nitrogen oxides from international maritime transport in European waters are projected to increase and could be equal to land-based sources by 2020 (EEA, 2013). " The reference dates back to 2013, do the authors have data on the current emissions? How accurate was EEA's projection from2013?

To our knowledge, there is as yet no current update on the land versus ship emissions in 2020 (this year). More recent projections with current emission control regulations, however, indicate that NOx emissions from international shipping will exceed those from land-based sources in the EU after 2030 (Cofala, 2018, http://pure.iiasa.ac.at/id/eprint/15729/).

- line 120: the scenarios "current legislation (CLE)" refer to the regulations included the lower sulfur limits from 2020 (see comment to line 67)? Or prior to 2020? Please clarify.

CLE 2020 refers to emissions according to the regulations in current legislation with the limits coming into force at the beginning of 2020.

- line 135, please review sentence (2 verbs): "The model results for 1990, 2000 and2010 were compared with the measurements available at the EDT project database based on EMEP datasets and model performance for SO2, NO2, PM10, PM2.5 and hourly O3 was discussed in detail in Jiang et al. (2020)."

We rephrased the sentence as follows:
P6, L179
*The model results for 1990, 2000 and 2010 were compared with the measurements available at the EDT project database which is based on EMEP datasets (https://wiki.met.no/emep/emep-experts/tfmmtrendstations). The number of available measurement stations varies between 15 and 64 depending on the year and species. For ozone, only measurements at the background-rural stations were used to reduce uncertainties due to the model resolution. Model performance for $SO_2$, $NO_2$, $PM_{10}$, $PM_{2.5}$ and hourly $O_3$ was discussed in detail in Jiang et al. (2020).*

- line 140: measuring ammonia is rather complex, therefore the quality of the observations should also be discussed (even if briefly). Please add some information on the measurement method and comparability between (the few) ammonia datasets available.

We provided detailed information about the measurements in a new table in the Supplementary (Table S1) and added the following statements in Section 3.1.:

P6, L188
*Atmospheric concentrations of ammonia are not well characterized due to relatively small number of monitoring sites, the short lifetime of $NH_3$ in the air and the difficulty of measuring non-point source emissions such as agricultural fields. Most of the measurement sites used in this study are located in the north; only very few stations are in the other parts of Europe (Fig. 1). The detailed information about the measurements (location, methods, temporal resolution) at each site is given in Table S1. Most of the measurements are daily concentrations, except for some sites in the Netherlands (hourly), Spain and Italy (weekly), Switzerland (bi-weekly) and UK (monthly). Measurement methods also differ; most of the stations use filter-pack sampling, while the passive samplers were used at 2 sites in Spain and the denuder systems were adopted at sites in the Netherlands, Great Britain and Switzerland. One should keep in mind that sampling artefacts due to the volatile nature of ammonium nitrate and the possible interaction with strong acids make separation of gases and particles by simple aerosol filters less reliable as indicated by EMEP (Co-operative Programme for Monitoring and Evaluation of the Long-Range Transmission of the Air*

*Pollutants in Europe), (https://projects.nilu.no/ccc/reports/cccr1-2019_Data_Report_2017.pdf). Modelled ammonia concentrations are similar to the measured ones at the few sites in the south while one site in eastern Europe shows an underestimation (Fig. 1). On the other hand, ammonia is overestimated at several sites such as in the UK, and in high emission areas around the Netherlands and Denmark. The mean fractional bias at all sites is 37.9% (Table S2).*

- line 156: "and/or deposition is underestimated by the model for which the resolution might also be critical factor". Is the model resolution not the same for all regions? If it is, then it could not explain the differences between central Europe and Iberian Peninsula and in Scandinavia (Fig. 1). Can the authors provide an explanation about why deposition might be more underestimated in central Europe than in the Iberian Peninsula and in Scandinavia? This seems a more likely cause for the model's overestimation around these high emission areas.

The horizontal resolution is the same everywhere in the model domain. The model performance for deposition, however, might differ in different regions in the domain due to various factors:
The spatial resolution used for the model simulations can add uncertainty to the model results, since the model estimate for a grid cell might not be representative of the specific location of the measurement site. Especially in mountainous areas with high spatial variability in precipitation patterns, spatial representativeness of the measurement sites is not possible. Furthermore, measurement sites close to farming areas may overestimate deposition of reduced nitrogen with respect to the modelled average deposition within the grid cell. Central Europe has more agricultural area and cattle farming than Scandinavia and the Iberian Peninsula. In addition, several studies show that the dry deposition velocity of ammonia (which is calculated using turbulent diffusion and surface characteristics in models) might vary significantly depending on the season and region (Flechard et al., 2011; Aksoyoglu and Prévôt, 2018). Different regional performance of the parameters used in the calculations might lead to different model performance for deposition. As reported by Theobald et al. (2019), the tendency of models to underestimate wet deposition and overestimate atmospheric concentrations is likely due to deficiencies in simulating wet-deposition processes, which are related to the vertical concentration profiles, scavenging coefficients or in-cloud processes, including the parameterisation of clouds.

We added the following paragraph in the revised text:

P7, L216
*These results suggest that ammonia emissions in the emission inventory might be too high around the main emission sources in central Europe and/or deposition is underestimated by the model for which the resolution might also be a critical factor. The model estimate for a grid cell might not be representative of the specific location of the measurement site. Especially in mountainous areas with very spatially variable precipitation patterns, spatial representativeness of the measurement sites is not possible. Furthermore, measurement sites close to farming areas may overestimate deposition of reduced nitrogen with respect to the modelled average deposition within the grid cell. In addition, several studies show that the dry deposition velocity of ammonia (which is calculated using turbulent diffusion and surface characteristics in*

*models) might vary significantly depending on the season and region (Flechard et al., 2011; Aksoyoglu and Prévôt, 2018). Therefore, different regional parameters used in the calculations might lead to different model performance for deposition. Moreover, as reported by Theobald et al. (2019), the tendency of models to underestimate wet deposition and overestimate atmospheric concentrations is likely due to deficiencies in simulating wet-deposition processes, which are related to the vertical concentration profiles, scavenging coefficients or in-cloud processes, including the parameterization of clouds.*

- line 171: "Among the SIA components, the best agreement between model and measurements is for sulfate". Can the authors quantify the relative difference (in %) between modelled and measured concentrations, for sulfate and nitrate respectively? It would be useful for the reader to have this information as well as the absolute difference (in microg/m3 shown in the Figure). Also for ammonia, in the previous paragraphs.

The relative difference between modelled and measured concentrations can be seen in Table S2 in the Supplement. For instance, mean fraction bias for sulfate is 4.7% while for nitrate it is 54.6%. For ammonia the MFB is 37.9%. We added this information from the supplement to the revised text as follows:

P7, L201
*On the other hand, ammonia is overestimated at several sites such as in the UK, and in high emission areas around the Netherlands and Denmark. The mean fractional bias at all sites is 37.9% (Table S2).*

P8, L245
*Among the SIA components, the best agreement between model and measurements is for sulfate (MFB = 4.7%) (Table S2, Fig. 1). The modeled concentrations of the other SIA components - for which the spatial coverage in central and western Europe is rather poor - are higher than the measured ones, especially for nitrate (MFB = 54.6%) (Fig. 1, Table S2).*

- lines 180-185: similarly to above, what is the reason for the increases in ammonia emissions? Is it increases in key sources (agriculture)? Or to a mix of emissions and atmospheric processes?
Please see the explanation above.

- line 197: "On the other hand, since simulations for 2030 were performed using the meteorological parameters of 2010, one should keep in mind that potentially higher temperatures in the future might increase the evaporation of ammonium nitrate to form its gaseous components NH3 and HNO3". This is a key point which could be high-lighted in the abstract.
Thank you for this suggestion. We added the following sentence in the abstract:

P2, L29
*One should also keep in mind that potentially higher temperatures in the future might increase the evaporation of ammonium nitrate to form its gaseous components $NH_3$ and $HNO_3$.*

- line 217: as above, please clarify what is meant by "current legislation" (before or after 2020): "Results of future scenario simulations suggest that sulfate concentrations will continue to decrease in central Europe as well as along shipping routes until 2030 assuming a current legislation (CLE) scenario (Fig. 3d, right panel)"

Current Legislation (CLE) is defined as legal and regulatory provisions in place at a certain agreed date. The ship emissions in 2020 and 2030 are projected based on current legislation (CLE) of the International Maritime Organization (IMO) and the EU. In such emission scenarios, it is assumed that emissions will be reduced by the amounts defined for 2020 and 2030 with respect to the reference year 2005.

- section 3.4: only as a suggestion, it might have been interesting to add an additional scenario including the implementation of a SECA in the Mediterranean (Rouïl,L., Ratsivalaka, C., André, J.-M., Allemand, N., 2019. ECAMED: a Technical Feasibility Study for the Implementation of an Emission Control Area (ECA) in the Mediterranean Sea. IMO report MEPC 74/INF.5.). An analysis of the potential impacts/benefits of this potential SECA in the framework of the authors' study could be very useful.

This is a very good suggestion. For a proper scenario calculation, however, the emission inventory has to be regenerated with modified ship emissions based on the SECA requirements. This can be done in a future project, but unfortunately not for this present study.
Moreover, similar scenario simulations have already been done. For example, Cofala et al., (2018) reported that designation of the Mediterranean Sea as an ECA could by 2030 cut emissions of $SO_2$ and $NO_x$ from international shipping by 80 and 20 percent, respectively, compared to current legislation.

- Table 1: please define the acronyms (CLE, MTFR) in the table header.
Done.

Aksoyoglu, S., and Prévôt, A. S. H.: Modelling nitrogen deposition: dry deposition velocities on various land-use types in Switzerland, Int. J. Environ. Pollut., Vol. 64, 230–245, https://doi.org/10.1504/IJEP.2018.10020573, 2018.

Flechard, C. R., Nemitz, E., Smith, R. I., Fowler, D., Vermeulen, A. T., Bleeker, A., Erisman, J. W., Simpson, D., Zhang, L., Tang, Y. S., and Sutton, M. A.: Dry deposition of reactive nitrogen to European ecosystems: a comparison of inferential models across the NitroEurope network, Atmos. Chem. Phys., 11 2703-2728, 10.5194/acp-11-2703-2011, 2011.

---

## Author Comment (AC2) · 4 Nov 2020

**Replies to the Anonymous Referee 2**

We would like to thank the referee for the valuable comments. Please find below our responses (in black) after the referee comments (in blue). Changes in the revised manuscript are written in *italics*.

This paper is a nice overview of the changing role of ammonia in the European atmosphere. I think it is generally well written, and my only major criticism is that a number of important uncertainties which might affect the conclusions are not discussed or addressed.

Major comments
The major areas of uncertainty that I missed include:
1. Bi-directionality. It is well established that ammonia can be emitted as well as deposited as a result of the equilibria between atmospheric and surface concentrations. It displays a so-called compensation point (Nemitz et al., 2001; Flechard et al., 1999, 2013), and this can affect the deposition close to source areas and long-range transport in general (Bash et al., 2013; Wichink Kruit et al., 2012).

Bi-directionality of ammonia is of course very important for the air concentration and dry deposition of ammonia.  Its treatment in the Zhang dry deposition algorithm was improved in the latest version of CAMx, which was released this year. We added the following statements in the new Section (2.2 Deposition Scheme):

P4, L111
*Dry and wet deposition of species were calculated using the Zhang scheme (Zhang et al., 2003; Ramboll, 2018). Although bi-directional air-surface exchange of $NH_3$ has been observed over a variety of land surfaces, most of the chemical transport models (CTMs) treat this exchange only as dry deposition that might lead to an underestimation of daytime $NH_3$ concentration because of overestimated dry deposition (Zhang et al., 2010). Winchink Kruit et al. (2012) reported that the inclusion of a stomatal compensation point led to increased modelled ammonia concentrations in agricultural areas in the Netherlands. Since stomatal compensation points are affected by the canopy type, temperature, growth stage, meteorological conditions, nitrogen status and cutting practices, it is very difficult to implement it in CTMs due to imprecise knowledge about the sub-grid variations in concentration, vegetation type and fertilizer applications (Huijsmans et al., 2018; Skjoth et al., 2011). Although the introduction of such a compensation point improves the model performance, the modelling of ammonia remains challenging due to temporal and spatial variations of emissions and grid resolution (Sutton et al., 2013). The bi-directional ammonia algorithm of Zhang et al. (2010) has been added recently as an option to the original Zhang deposition algorithm in the latest version of CAMx (v7.00). Default landuse-dependent emission potentials control ammonia compensation points along the surface-air transport circuit.  When the atmospheric ammonia concentration exceeds the compensation point, the net flux is from air to surface; in the opposite case, the net flux is from surface to air. Although the Zhang dry deposition algorithm in the previous version of the CAMx (v6.50) model used in this study did not include compensation points, it did treat bi-directionality indirectly by using a deposition parameter that strongly influenced ammonia deposition via surface resistance.*

2. This study seems to ignore the impacts of co-deposition, in which the acidity of the surface (affected by both SO2 and NH3 emissions, and their trends) changes. The impacts of this process on trends have been explored in for example Wichink Kruit et al. (2017).

The CAMx dry deposition model considers these effects. We added the following text in the revised manuscript:

P5, L129

3. A similar issue with trends, also not mentioned, is changing pH of rainwater (Banzhaf et al., 2012).
The pH-dependent parameterizations are incorporated and cloud water pH is calculated by the aqueous-phase chemistry algorithms in the CAMx model. We added this information in the text as follows:

P5, L138
*Wet deposition is the predominant removal process for fine particles. Particles act as cloud condensation nuclei and the resulting cloud droplets grow into precipitation. The CAMx wet deposition model employs a scavenging approach using the 3-dimensional cloud/rain input from the meteorological model. Banzaf et al. (2012) reported that droplet pH variation within atmospheric ranges affects modelled air concentrations and wet deposition fluxes significantly. The pH-dependent parameterizations are incorporated and cloud water pH is calculated by the aqueous-phase chemistry algorithms in the CAMx model.*

4. Meteorological variability. The current study mainly uses meteorology from just two years, 1990 and 2010, but Wichink Kruit et al. (2017) showed that meteorology can also account for a significant contribution to NH3 trends.
We would like to emphasize that our aim in this study was not to calculate trends for which continuous, long-term simulations are required (continuous simulation of 21 years between 1990 and 2010), as already done in the EDT project (Colette et al., 2017; Ciarelli et al., 2019). In this study, however, we performed the simulations for the 3 base years in the past 1990, 2000 and 2010 using the meteorology of each of those 3 years. We used the meteorology of 2010 only for the future scenarios and discussed the potential effects of different future meteorology in the text. The effect of meteorology on ammonia is well known. Both emissions and chemistry (particulate nitrate formation) are affected by meteorological conditions – mainly temperature. Backes et al. (2016) and Hendriks et al., (2016) showed that the modelling of ammonia concentrations can be improved when ammonia emissions are modulated by local meteorological conditions. The trends calculated by several models for the full 21 years between 1990-2010 were analyzed and compared to the observed trends during the Eurodelta-Trends project (Ciarelli et al., 2019). Therefore, the models in the EDT study were able to take into account the effect of meteorology on chemistry, but not the effect of temperature on ammonia emissions; these were based on static emission profiles. We expanded the section 3.1 as follows:

P6, L188
*Atmospheric concentrations of ammonia are not well characterized due to relatively small number of monitoring sites, the short lifetime of NH3 in the air and the difficulty*

*of measuring non-point source emissions such as agricultural fields. Most of the measurement sites used in this study are located in the north; only very few stations are in the other parts of Europe (Fig. 1). The detailed information about the measurements (location, methods, temporal resolution) at each site is given in Table S1. Most of the measurements are daily concentrations, except for some sites in the Netherlands (hourly), Spain and Italy (weekly), Switzerland (bi-weekly) and UK (monthly). Measurement methods also differ; most of the stations use filter-pack sampling, while the passive samplers were used at 2 sites in Spain and the denuder systems were adopted at sites in the Netherlands, Great Britain and Switzerland. One should keep in mind that sampling artefacts due to the volatile nature of ammonium nitrate and the possible interaction with strong acids make separation of gases and particles by simple aerosol filters less reliable as indicated by EMEP (Co-operative Programme for Monitoring and Evaluation of the Long-Range Transmission of the Air Pollutants in Europe), (https://projects.nilu.no/ccc/reports/cccr1-2019_Data_Report_2017.pdf). Modelled ammonia concentrations are similar to the measured ones at the few sites in the south while one site in eastern Europe shows an underestimation (Fig. 1). On the other hand, ammonia is overestimated at several sites such as in the UK, and in high emission areas around the Netherlands and Denmark. The mean fractional bias at all sites is 37.9% (Table S2). Overestimation might originate from either overestimated emissions or underestimated removal (deposition, particle formation). There are still large uncertainties about ammonia emissions. Recent studies show that better agreement between models and measurements can be achieved when ammonia emissions are modulated with local meteorological conditions (Backes et al., 2016; Hendriks et al., 2016). Most models, however rely on the static ammonia emission profiles provided in the emission inventories (Ciarelli et al., 2019).*

5. Ship-plumes. It is well known that models tend to mis-represent HNO3 production from NO emitted from ships into clean marine environments (von Glasow et al., 2003; Vinken et al., 2011, 2014). This could potentially have been handled with the CAMx model'splume-in-grid approach, but this doesn't seem to have been used. However, some of the comments made about HNO3 (e.g. L187 onwards) may be impacted by this issue.

The Plume-in-Grid (PiG) sub-model in CAMx addresses the size and chemical evolution of point source plumes and is used for stationary sources such as power plants. Using PiG for ship emissions would require modelling the plumes from each of different, individual emission sources, which would be computationally impossible. The main obstacle, however, arises from the fact that these sources (ships) are moving. As Vinken et al. (2011) showed, accounting for in-plume chemistry is most relevant for pristine marine environments. We believe that the effect of plume-in-grid non-linear chemistry is very small in polluted areas with heavy ship traffic along the European coastal areas and other uncertainties coming mainly from emissions are more important. In this study, the ship emissions were not injected into the first model layer as ground emissions, but into the second layer. We added the following paragraph in the Methods Section:

P5, L157

*The anthropogenic emissions were distributed to various vertical layers depending on their sources using the vertical profile given by Bieser et al. (2011). The ship emissions*

*over the sea were injected into the second model layer. All the biogenic emissions were released into the surface layer.*

6. In the introduction, I missed some discussion of trend studies on land-based emissions and concentration/deposition trends which have already been done, e.g. Fowler et al.(2007); Fagerli and Aas (2008) or Wichink Kruit et al. (2017). How does the current study add to these? (Page 3 gives a lot of information given on the impacts of shipping, but not much about land.)

We extended the part about the emission reductions in the Introduction as follows:

P3, L48
*European anthropogenic emissions have decreased substantially since the 1990s as a result of large emission reductions following the Gothenburg Protocol (GP) (UNECE, 1999), revised Gothenburg Protocol (revised on 4 May 2012, [https://www.unece.org/env/lrtap/multi_h1.html](https://www.unece.org/env/lrtap/multi_h1.html)) and EU Directives ([https://www.eea.europa.eu/data-and-maps/indicators/main-anthropogenic-air-pollutant-emissions/assessment-6](https://www.eea.europa.eu/data-and-maps/indicators/main-anthropogenic-air-pollutant-emissions/assessment-6)). Several studies investigated the effects of reduced land emissions on the air quality in various parts of Europe (Guerreiro et al., 2014, Aksoyoglu et al., 2014; Wichink Kruik et al., 2017; van Zanten et al., 2017; Theobald et al., 2019; Ciarelli et al., 2019). The largest decrease was in $SO_2$ emissions (by more than 90% in 2017 compared to 1990), followed by $NO_x$ and NMVOC (non-methane volatile organic compounds) emission reductions (more than 50%), while ammonia emissions decreased less – approximately 23% on average in the EU-28 countries. Ammonia emissions have been increasing again since 2014, however, posing problems for Europe (NEC, 2019). This is mainly due to the difficulty in implementing additional emission reductions in the agriculture sector, especially in the housing of animals and the storage and application of animal manures. The large decrease in sulfur emissions over the last few decades has changed the aerosol composition: particulate nitrogen was dominated by sulfates in the 1990s while today nitrate predominates (Colette et al., 2016).*

We replaced the reference by Fowler et al., (2009; 2015) in P2, L37.

2. L37. The authors only mention ammonium sulfate here, but bi-sulfate is an important component of European aerosol too.

We modified the relevant paragraph as:

P2, L38
*Ammonia reacts very rapidly with sulfuric acid ($H_2SO_4$), which is formed from the oxidation of $SO_2$ by OH in the gas phase and by $O_3$, hydrogen peroxide ($H_2O_2$) and other oxidants in the aqueous phase, to form ammonium sulfate (($NH_4$)$_2SO_4$) or ammonium bisulfate ($NH_4HSO_4$) (Seinfeld and Pandis, 2012). If there is enough ammonia available after the neutralization of $H_2SO_4$, it reacts with nitric acid ($HNO_3$) to produce ammonium nitrate. These secondary inorganic aerosols (SIA) contribute*

*most to the fine particulate matter ($PM_{2.5}$) in Europe (Ciarelli et al., 2016; 2019; Aksoyoglu et al, 2017).*

3. L42. The Dentener ref is 14 years old now; find something more

Dentener et al., (2006) was replaced by Jones et al. (2014) in P2, L45

4. L53. Are you sure it is ammonium sulfate, and not bisulfate?

A series of compounds may exist in the aerosol phase like $(NH_4)_3H(SO_4)_2$, $(NH_4)_2SO_4$ and $NH_4HSO_4$ depending on the availability of ammonia, sulfuric acid, temperature and relative humidity. Bisulfate exists in acidic atmospheres with low ammonia availability. The ISORROPIA model in CAMx deals with the inorganic aerosol thermodynamics/partitioning. In order to avoid confusion, we rephrased the sentence in the text as:

*P3, L59*
*The large decrease in sulfur emissions over the last few decades has changed the aerosol composition: particulate nitrogen was dominated by sulfates in the 1990s while today nitrate predominates (Colette et al., 2016).*

5. L54. The Colette et al 2016 reference seems to be some grey literature, with no address and no url. What is this? And surely there are some peer-reviewed papers that be cited to support this statement?
It is an EMEP Report. We added the complete citation: EMEP: CCCP Report 1/2016, https://projects.nilu.no/ccc/reports/cccr1-2016.pdf, NILU, Oslo, 2016

6. L65-17. The SECA's came into effect at the start of 2015
SECAs were introduced in Europe in July 2010 with sulfur limit of 1.0% and then it was further reduced to 0.1% in January 2015. We modified the sentence as follows:

P3, L73
*For example, in Europe, the North Sea and Baltic Sea areas were defined as SECAs (sulfur emission control areas), where the limits were restricted to 1.0% in July 2010 and further reduced to 0.1% in 1 January 2015. New global sulfur emission regulations, which reduce limits from 3.5% to 0.5% came into force on 1 January 2020 (https://www.imo.org/en/MediaCentre/HotTopics/Pages/Sulphur-2020.aspx, last access on 23.10.2020).*

7. L80-84. It is unclear where the cited 1-14% PM2.5 applies. This number sounds very different to those cited for Karl et al., and so this paragraph is a little confusing. Are the Karl et al results similar to, or very different from those cited from Viana et al.?
Viana et al. is a literature review of past studies (until 2012) where different calculation methods were used at different locations. The numbers given in that paper therefore, vary depending on location in Europe and the time period the studies were performed. On the other hand, Karl et al. is a model intercomparison study for 2012 only for the Baltic Sea. To avoid confusion, we modified the paragraph as follows:

P3, L84

*Viana et al. (2014) reviewed a series of studies performed before 2012 dealing with the impact of shipping emissions on air quality in the European coastal areas and reported that contribution of ship emissions to PM$_{2.5}$ and to NO$_2$ vary between 1-14% and 7-24%, respectively, depending on location and time. In a recent model-intercomparison study, Karl et al. (2019) evaluated the contribution of ship emissions to air quality in the Baltic Sea region in 2012 to investigate the differences among model predictions and showed that variations in ship-related PM$_{2:5}$ were mainly due to differences in the models' schemes for inorganic aerosol formation.*

8. L100-102 How is the coarse-mode aerosol (e.g. for nitrate) handled in this model system?
The coarse-mode nitrate is treated in the coarse fraction (PM10-PM2.5) in CAMx. In this study, we only investigated the fine aerosol (PM2.5).

9. L103. Be explicit with a reference to the Zhang scheme (not just the cited CAMx user's guide). And whether co-deposition is included or not?

We included the original reference Zhang et al. (2003) in the revised manuscript (Section 2.2) for the Zhang scheme (P4, L112)

10. L134 on. Brief details on the measurement networks underlying EDT work should be given.
We added some information about the measurement networks used for model evaluation in Section 3.1:

P6, L179
*The model results for 1990, 2000 and 2010 were compared with the measurements available at the EDT project database which is based on EMEP datasets (https://wiki.met.no/emep/emep-experts/tfmmtrendstations). The number of available measurement stations varies between 15 and 64 depending on the year and species. For ozone, only measurements at the background-rural stations were used to reduce uncertainties due to the model resolution. Model performance for SO$_2$, NO$_2$, PM$_{10}$, PM$_{2.5}$ and hourly O$_3$ was discussed in detail in Jiang et al. (2020).*

11. L140 I won't repeat Ref #1's comments, but agree with them.
We provided detailed information about the measurements in a new table in the Supplementary (Table S1) and added the following statements in Section 3.1.:

P6, L188
*Atmospheric concentrations of ammonia are not well characterized due to relatively small number of monitoring sites, the short lifetime of NH$_3$ in the air and the difficulty of measuring non-point source emissions such as agricultural fields. Most of the measurement sites used in this study are located in the north; only very few stations are in the other parts of Europe (Fig. 1). The detailed information about the measurements (location, methods, temporal resolution) at each site is given in Table S1. Most of the measurements are daily concentrations, except for some sites in the Netherlands (hourly), Spain and Italy (weekly), Switzerland (bi-weekly) and UK (monthly). Measurement methods also differ; most of the stations use filter-pack sampling, while the passive samplers were used at 2 sites in Spain and the denuder*

*systems were adopted at sites in the Netherlands, Great Britain and Switzerland. One should keep in mind that sampling artefacts due to the volatile nature of ammonium nitrate and the possible interaction with strong acids make separation of gases and particles by simple aerosol filters less reliable as indicated by EMEP (Co-operative Programme for Monitoring and Evaluation of the Long-Range Transmission of the Air Pollutants in Europe), (https://projects.nilu.no/ccc/reports/cccr1-2019_Data_Report_2017.pdf). Modelled ammonia concentrations are similar to the measured ones at the few sites in the south while one site in eastern Europe shows an underestimation (Fig. 1). On the other hand, ammonia is overestimated at several sites such as in the UK, and in high emission areas around the Netherlands and Denmark. The mean fractional bias at all sites is 37.9% (Table S2). Overestimation might originate from either overestimated emissions or underestimated removal (deposition, particle formation). There are still large uncertainties about ammonia emissions. Recent studies show that better agreement between models and measurements can be achieved when ammonia emissions are modulated with local meteorological conditions (Backes et al., 2016; Hendriks et al., 2016). Most models, however rely on the static ammonia emission profiles provided in the emission inventories (Ciarelli et al., 2019).*

12. L145. As noted above, many processes not discussed in this manuscript might also contribute to model-measurement bias. Another issues is scale, which is very briefly mentioned on L156, but which can be a very important factor for NH3 (Theobald et al.,2016; Denby et al., 2020).

There are of course several factors which could contribute to the bias. The most important one, however, is the large uncertainty in the quantity as well as temporal variation of ammonia emissions.

13. Notation. Better to use pNH4, pNO3, pSO4 than PNH4 etc, to avoid mixing chemical and atmospheric nomenclature.

The aerosol components are defined with capital "P" in the model and used in all the publications in the same way. We prefer to keep them as they are in order to be consistent with our previous publications.

14. L242 states that the amount of precipitation is crucial, but no figures are given on this here; please expand.

Precipitation is calculated by the meteorological model. In this study, the meteorological input was obtained from the Eurodelta-Trends project as described in Jiang et al. (2020). The performance evaluation of the seasonal and annual accumulated precipitation used in the Eurodelta-Trends exercise is discussed in detail in Theobald et al. (2019). We expanded this part in Section 3.3.2:

P10, L317
*The performance evaluation of the accumulated precipitation used in the Eurodelta-Trends exercise is discussed in detail in Theobald et al. (2019). The model biases are very small for accumulated annual precipitation for the meteorological model used in this study; there is an underestimation of 4%-8%.*

15. Many of the figure legends and colors need to be re-done. For example, in Fig2c reds are used for positive values and blues for negative, which is great, but in Figs. 2d

and 2e the color-scale shows white for levels both above and below certain thresholds! Later Figures also show such strange behaviors. I suggest using the same color-scale for all subplots, and do not have the same color for different values.

The reviewer probably means Fig.2b versus Figs. 2c and 2d, since there is no Fig. 2e in the manuscript. In all the difference plots (change between years), the same color scheme was used, i.e. no change (around zero) is always white, positive values (increase) are always red and negative values (decrease) are always blue. Since the scales are very different for different species (more than a factor of 10), it is not possible to use the same color for the same number. We did, however, keep the same scale for the same species in different time periods (e.g. always from -6 to +3 ppb for $NH_3$ in Fig2b-d (left panels) and always from -0.1 to +0.4 ppb for $HNO_3$ in Fig2b-d (right panels).

16. Fig. S2 - which measurements? Be explicit.
The description of the measurements used for model evaluation is given in Section 3.1. We added the relevant information in the caption of FigS2 as follows:

*Measurements are from the EMEP network (see Section 3.1 in the main text).*

References
Banzhaf, S., Schaap, M., Kerschbaumer, A., Reimer, E., Stern, R., van der Swaluw, E., and Builtjes, P.: Implementation and evaluation of pH-dependent cloud chemistry and wet deposition in the chemical transport model REM-Calgrid, Atmos. Environ., 49, 378 390, https://doi.org/10.1016/j.atmosenv.2011.10.069, 2012.
Bash, J. O., Cooter, E. J., Dennis, R. L., Walker, J. T., and Pleim, J. E.: Evaluation of a regional air-quality model with bidirectional NH3 exchange coupled to an agroecosystem model, Biogeosciences, 10, 1635–1645, https://doi.org/10.5194/bg-10-1635-2013, http://www.biogeosciences.net/10/1635/2013/, 2013.
Denby, B., Gauss, M., Wind, P., Mu, Q., Grøtting Wærsted, E., Fagerli, H., Valdebenito,A., and Klein, H.: Description of the uEMEPv5 downscaling approach for the EMEPMSC-W chemistry transport model, Geoscientific Model Development Discussions, 2020, 1–38, https://doi.org/10.5194/gmd-2020-119,https://gmd.copernicus.org/preprints/gmd-2020-119/, 2020.
Fagerli, H. and Aas, W.: Trends of nitrogen in air and precipitation: Model results and observations at EMEP sites in Europe, 1980-2003, Environ. Poll., 154, 448–461, 2008.
Flechard, C. R., Fowler, D., Sutton, M. A., and Cape, J. N.: A dynamic chemical model of bi-directional ammonia exchange between semi-natural vegetation and the atmosphere, Q.J. R. Meteorol. Soc., 125, 2611–2641, 1999.
Flechard, C. R., Massad, R.-S., Loubet, B., Personne, E., Simpson, D., Bash, J. O., Cooter,E. J., Nemitz, E., and Sutton, M. A.: Advances in understanding, models and parameterizations of biosphere-atmosphere ammonia exchange, Biogeosciences, 10, 5183–5225, https://doi.org/10.5194/bg-10-5183-2013, http://www.biogeosciences.net/10/5183/2013/, 2013.
Fowler, D., Smith, R., Muller, J., Cape, J., Sutton, M., Erisman, J., and Fagerli, H.: Long Term Trends in Sulphur and Nitrogen Deposition in Europe and the Cause of Non-linearities, Water, Air, & Soil Pollution: Focus, 7, 41–47,http://dx.doi.org/10.1007/s11267-006-9102-x, 2007.
Nemitz, E., Milford, C., and Sutton, M. A.: A two-level canopy compensation point model for describing bi-directional biosphere-atmosphere exchange of ammonia,Q. J. R. Meteorol.Soc., 127, 815–833, 2001.

Theobald, M. R., Simpson, D., and Vieno, M.: Improving the spatial resolution of air-quality modelling at a European scale – development and evaluation of the Air Quality Re-gridder Model (AQR v1.1), Geoscientific Model Development, 9, 4475–4489, https://doi.org/10.5194/gmd-9-4475-2016, http://www.geosci-model-dev.net/9/4475/2016/, 2016. Vinken, G. C. M., Boersma, K. F., Jacob, D. J., and Meijer, E. W.: Accounting for non-linear chemistry of ship plumes in the GEOS-Chem global chemistry transport model, Atmospheric Chemistry and Physics, 11, 11 707–11 722, https://doi.org/10.5194/acp-11-11707-2011, http://www.atmos-chem-phys.net/11/11707/2011/, 2011.

Vinken, G. C. M., Boersma, K. F., van Donkelaar, A., and Zhang, L.: Constraints on ship NOx emissions in Europe using GEOS-Chem and OMI satellite NO2 observations, Atmospheric Chemistry and Physics, 14, 1353–1369, https://doi.org/10.5194/acp-14-1353-2014, http://www.atmos-chem-phys.net/14/1353/2014/, 2014.

von Glasow, R., Lawrence, M. G., Sander, R., and Crutzen, P. J.: Modeling the chemical effects of ship exhaust in the cloud-free marine boundary layer, Atmos. Chem. Physics, 3,233–250, http://www.atmos-chem-phys.net/3/233/2003/, 2003.

Wichink Kruit, R. J., Schaap, M., Sauter, F. J., van Zanten, M. C., and van Pul, W.A. J.: Modeling the distribution of ammonia across Europe including bi-directional surface-atmosphere exchange, Biogeosciences, 9, 5261–5277, https://doi.org/10.5194/bg-9-5261-2012, 2012.

Wichink Kruit, R. J., Aben, J., de Vries, W., Sauter, F., van der Swaluw, E., van Zanten,M. C., and van Pul, W. A. J.: Modelling trends in ammonia in the Netherlands over the1990-2014, Atmos. Environ., 154, 20–30, https://doi.org/10.1016/j.atmosenv.2017.01.031,2017.

---

## Author Comment (AC3) · 5 Nov 2020

We noticed that one of our replies was not fully implemented in the uploaded file. Please find below the complete answer with the missing part :

reviewer comment:2. This study seems to ignore the impacts of co-deposition, in which the acidity of the surface (affected by both SO2 and NH3 emissions, and their trends) changes. The impacts of this process on trends have been explored in for example Wichink Kruit et al. (2017).

our reply: The CAMx dry deposition model considers these effects. We added the

following text in the revised manuscript:

P5, L129 The surface resistance is an area of great uncertainty in deposition models. Surface wetness plays an important role for both cuticular and ground resistance. This effect is included in some deposition velocity algorithms. The parameterizations for wet cuticles and ground are quite variable between models. Some models such as CMAQ use the Henry's Law constant to account for the solubility of chemical species, the EMEP model (Simpson et al., 2012) considers the chemical content of dew by treating co-deposition of species such as SO2 and NH3 while Zhang et al. (2003) includes the consideration of friction velocity. Wichink Kruit et al. (2017) showed the effects of co-deposition, chemistry and meteorology during 1993-2014 in the Netherlands. Relatively wet conditions led to lower ammonia concentrations, while warm and dry conditions led to higher levels.